# Identification of autosomal cis expression quantitative trait methylation (cis eQTMs) in children's blood

Carlos Ruiz-Arenas[1,2]\*, Carles Hernandez-Ferrer[2,3,4], Marta Vives-Usano[2,4,5], Sergi Marí[2,4,6], Ines Quintela[7], Dan Mason[8], Solène Cadiou[9], Maribel Casas[2,4], Sandra Andrusaityte[10], Kristine Bjerve Gutzkow[11], Marina Vafeiadi[2,4,12], John Wright[8], Johanna Lepeule[9], Regina Grazuleviciene[10], Leda Chatzi[13], Ángel Carracedo[14,15], Xavier Estivill[16], Eulàlia Marti[6,17], Geòrgia Escaramís[6,17], Martine Vrijheid[2,4,6], Juan R González[2,4,6], Mariona Bustamante[2,4,6]\*

[1]Centro de Investigación Biomédica en Red de Enfermedades Raras (CIBERER), Barcelona, Spain; [2]Universitat Pompeu Fabra (UPF), Barcelona, Spain; [3]Centro Nacional de Análisis Genómico (CNAG-CRG), Center for Genomic Regulation, Barcelona Institute of Science and Technology (BIST), Barcelona, Spain; [4]ISGlobal, Barcelona, Spain; [5]Center for Genomic Regulation (CRG), Barcelona Institute of Science and Technology, Barcelona, Spain; [6]CIBER Epidemiología y Salud Pública (CIBERESP), Barcelona, Spain; [7]Medicine Genomics Group, University of Santiago de Compostela, Santiago de Compostela, Spain; [8]Bradford Institute for Health Research, Bradford Teaching Hospitals NHS Foundation Trust, Bradford, United Kingdom; [9]University Grenoble Alpes, Inserm, CNRS, Team of Environmental Epidemiology Applied to Reproduction and Respiratory Health, Grenoble, France; [10]Department of Environmental Science, Vytautas Magnus University, Kaunas, Lithuania; [11]Department of Environmental Health, Norwegian Institute of Public Health, Oslo, Norway; [12]Department of Social Medicine, University of Crete, Crete, Greece; [13]Department of Preventive Medicine, Keck School of Medicine, University of Southern California, Los Angeles, Los Angeles, United States; [14]Medicine Genomics Group, CIBERER, University of Santiago de Compostela, Santiago de Compostela, Spain; [15]Galician Foundation of Genomic Medicine, Santiago de Compostela, Spain; [16]Quantitative Genomics Medicine Laboratories (qGenomics), Esplugues del Llobrega, Barcelona, Spain; [17]Departament de Biomedicina, Institut de Neurociències, Universitat de Barcelona, Barcelona, Spain

\*For correspondence:
carlos.ruiza@upf.edu (CR-A);
mariona.bustamante@isglobal.org (MB)

Competing interest: The authors declare that no competing interests exist.

## Abstract

**Background:** The identification of expression quantitative trait methylation (eQTMs), defined as associations between DNA methylation levels and gene expression, might help the biological interpretation of epigenome-wide association studies (EWAS). We aimed to identify autosomal cis eQTMs in children's blood, using data from 832 children of the Human Early Life Exposome (HELIX) project.

**Methods:** Blood DNA methylation and gene expression were measured with the Illumina 450 K and the Affymetrix HTA v2 arrays, respectively. The relationship between methylation levels and expression of nearby genes (1 Mb window centered at the transcription start site, TSS) was assessed by fitting 13.6 M linear regressions adjusting for sex, age, cohort, and blood cell composition.

**Results:** We identified 39,749 blood autosomal cis eQTMs, representing 21,966 unique CpGs (eCpGs, 5.7% of total CpGs) and 8,886 unique transcript clusters (eGenes, 15.3% of total transcript clusters, equivalent to genes). In 87.9% of these cis eQTMs, the eCpG was located at <250 kb from eGene's TSS; and 58.8% of all eQTMs showed an inverse relationship between the methylation and expression levels. Only around half of the autosomal cis-eQTMs eGenes could be captured through annotation of the eCpG to the closest gene. eCpGs had less measurement error and were enriched for active blood regulatory regions and for CpGs reported to be associated with environmental exposures or phenotypic traits. In 40.4% of the eQTMs, the CpG and the eGene were both associated with at least one genetic variant. The overlap of autosomal cis eQTMs in children's blood with those described in adults was small (13.8%), and age-shared cis eQTMs tended to be proximal to the TSS and enriched for genetic variants.

**Conclusions:** This catalogue of autosomal cis eQTMs in children's blood can help the biological interpretation of EWAS findings and is publicly available at https://helixomics.isglobal.org/ and at Dryad (doi:10.5061/dryad.fxpnvx0t0).

**Funding:** The study has received funding from the European Community's Seventh Framework Programme (FP7/2007-206) under grant agreement no 308333 (HELIX project); the H2020-EU.3.1.2. - Preventing Disease Programme under grant agreement no 874583 (ATHLETE project); from the European Union's Horizon 2020 research and innovation programme under grant agreement no 733206 (LIFECYCLE project), and from the European Joint Programming Initiative "A Healthy Diet for a Healthy Life" (JPI HDHL and Instituto de Salud Carlos III) under the grant agreement no AC18/00006 (NutriPROGRAM project). The genotyping was supported by the projects PI17/01225 and PI17/01935, funded by the Instituto de Salud Carlos III and co-funded by European Union (ERDF, "A way to make Europe") and the Centro Nacional de Genotipado-CEGEN (PRB2-ISCIII). BiB received core infrastructure funding from the Wellcome Trust (WT101597MA) and a joint grant from the UK Medical Research Council (MRC) and Economic and Social Science Research Council (ESRC) (MR/N024397/1). INMA data collections were supported by grants from the Instituto de Salud Carlos III, CIBERESP, and the Generalitat de Catalunya-CIRIT. KANC was funded by the grant of the Lithuanian Agency for Science Innovation and Technology (6-04-2014_31V-66). The Norwegian Mother, Father and Child Cohort Study is supported by the Norwegian Ministry of Health and Care Services and the Ministry of Education and Research. The Rhea project was financially supported by European projects (EU FP6-2003-Food-3-NewGeneris, EU FP6. STREP Hiwate, EU FP7 ENV.2007.1.2.2.2. Project No 211250 Escape, EU FP7-2008-ENV-1.2.1.4 Envirogenomarkers, EU FP7-HEALTH-2009- single stage CHICOS, EU FP7 ENV.2008.1.2.1.6. Proposal No 226285 ENRIECO, EU- FP7- HEALTH-2012 Proposal No 308333 HELIX), and the Greek Ministry of Health (Program of Prevention of obesity and neurodevelopmental disorders in preschool children, in Heraklion district, Crete, Greece: 2011-2014; "Rhea Plus": Primary Prevention Program of Environmental Risk Factors for Reproductive Health, and Child Health: 2012-15). We acknowledge support from the Spanish Ministry of Science and Innovation through the "Centro de Excelencia Severo Ochoa 2019-2023" Program (CEX2018-000806-S), and support from the Generalitat de Catalunya through the CERCA Program. MV-U and CR-A were supported by a FI fellowship from the Catalan Government (FI-DGR 2015 and #016FI_B 00272). MC received funding from Instituto Carlos III (Ministry of Economy and Competitiveness) (CD12/00563 and MS16/00128).

## Editor's evaluation

The study investigates, for the first time, gene regulation by DNA methylation across the genome in children. The results will be useful for better interpreting the many completed and ongoing studies of the effects of environmental and disease on DNA methylation in children. Prior to this study, investigators had to make do with inaccurate information derived from adult studies.

## Introduction

Cells from the same individual, although sharing the same genome sequence, differentiate into diverse lineages that finally give place to specific cell types with unique functions. This is orchestrated by the epigenome, which regulates gene expression in a cell/tissue- and time-specific manner (*Cavalli and*

**eLife digest** Cells can fine-tune which genes they activate, when and at which levels using a range of chemical marks on the DNA and certain proteins that help to organise the genome. One well-known example of such 'epigenetic tags' is DNA methylation, whereby a methyl group is added onto particular positions in the genome. Many factors – including environmental effects such as diet – control DNA methylation, allowing an organism to adapt to ever-changing conditions.

An expression quantitative trait methylation (eQTM) is a specific position of the genome whose DNA methylation status regulates the activity of a given gene. A catalogue of eQTMs would be useful in helping to reveal how the environment and disease impacts the way cells work. Yet, currently, the relationships between most epigenetic tags and gene activity remains unclear, especially in children.

To fill this gap, Ruiz-Arenas et al. studied DNA methylation in blood samples from over 800 healthy children across Europe. Amongst all tested DNA methylation sites, 22,000 (5.7% of total) were associated with the expression of a gene – and therefore were eQTMs; reciprocally, 9,000 genes (15.3% of all tested genes) were linked to at least one methylation site, leading to a total of 40,000 pairs of DNA methylation sites and genes. Most often, eQTMs regulated the expression of nearby genes – but only half controlled the gene that was the closest to them. Age and the genetic background of the individuals influenced the nature of eQTMs.

This catalogue is a useful resource for the scientific community to start understanding the relationship between epigenetics and gene activity. Similar studies are now needed for other tissues and age ranges. Overall, extending our knowledge of eQTMs may help reveal how life events lead to illness, and could inform prevention efforts.

*Heard, 2019*; *Feinberg, 2018*; *Lappalainen and Greally, 2017*). Besides its central role in regulating embryonic and fetal development, X-chromosome inactivation, genomic imprinting, and silencing of repetitive DNA elements, the epigenome is also responsible for the plasticity and cellular memory in response to environmental perturbations (*Cavalli and Heard, 2019*; *Feinberg, 2018*; *Lappalainen and Greally, 2017*).

Massive epigenetic alterations, caused by somatic mutations, age, injury, or environmental exposures, were initially described in cancer (*Feinberg, 2018*). The paradigm of environmental factors modifying the epigenome and leading to increased disease risk was then extrapolated from cancer to a wide range of common diseases. Consequently, in recent years, a high number of epigenome-wide association studies (EWAS) have been performed, investigating the relation of prenatal and postnatal exposure to environmental factors with DNA methylation, and of DNA methylation with disease (*Feinberg, 2018*; *Lappalainen and Greally, 2017*). EWAS findings have been inventoried in two catalogues: the EWAS catalog (*Battram et al., 2021*) and the EWAS Atlas (*Li et al., 2019*). The latter includes 0.5 M associations for 498 traits from 1216 studies, including 155 different cells/tissues.

Despite the success of EWAS in identifying altered methylation patterns, various challenging issues still must be solved: the role of genetic variation; the access to the target tissue/cell; confounding reverse causation; and biological interpretation (*Feinberg, 2018*; *Lappalainen and Greally, 2017*). Regarding the latter, most studies do not have transcriptional data to test the effect of DNA methylation on gene expression. When these data are not available, a common approach is to assume that CpG DNA methylation affects the expression of the closest gene (*Sharp et al., 2017*). Although this approach is easy to implement, it is limited. Indeed, CpG DNA methylation might regulate distant genes or might not regulate any gene at all (*Bonder et al., 2017*; *Lappalainen and Greally, 2017*). Another approach to elucidate the effect of DNA methylation on gene expression when transcriptional data are not available is to rely on previous expression quantitative trait methylation (eQTM) studies. These are genome-wide studies investigating the associations between the levels of DNA methylation and gene expression (*Gondalia et al., 2019*; *Küpers et al., 2019*). Several eQTM studies have been performed in diverse cell types/tissues: whole blood (*Bonder et al., 2017*; *Kennedy et al., 2018*), monocytes (*Husquin et al., 2018*; *Kennedy et al., 2018*; *Liu et al., 2013*), lymphoblastoid cell lines, T-cells and fibroblasts derived from umbilical cords (*Gutierrez-Arcelus et al., 2015*; *Gutierrez-Arcelus et al., 2013*), fibroblasts (*Wagner et al., 2014*), liver (*Bonder et al., 2014*), skeletal muscle (*Taylor et al., 2019*), nasal airway epithelium (*Kim et al., 2020*), and placenta (*Delahaye et al., 2018*).

As most of the EWAS are conducted in whole blood (*Felix et al., 2018*; *Li et al., 2019*), there is a need for comprehensive eQTM studies in this tissue. To date, available eQTM studies in whole blood only cover samples from adults (*Bonder et al., 2017*; *Kennedy et al., 2018*) and their validity in children has not been assessed.

In this study, we analyzed DNA methylation and gene expression data from the Human Early-Life Exposome (HELIX) project to generate an autosomal cis eQTM catalogue in children's blood (https://helixomics.isglobal.org/). We analyzed the proportion of cis eQTMs captured through annotation to the closest gene, characterized them at the functional level, assessed the influence of genetic variation and compared them with eQTMs identified in adults. An overview of all the analyses can be found in *Figure 1*. This public resource will help the functional interpretation of EWAS findings in children.

## Methods

### Sample of the study

The Human Early Life Exposome (HELIX) study is a collaborative project across six established and on-going longitudinal population-based birth cohort studies in Europe (*Maitre et al., 2018*): the Born in Bradford (BiB) study in the UK (*Wright et al., 2013*), the Étude des Déterminants pré et postnatals du développement et de la santé de l'Enfant (EDEN) study in France (*Heude et al., 2016*), the INfancia y Medio Ambiente (INMA) cohort in Spain (*Guxens et al., 2012*), the Kaunus cohort (KANC) in Lithuania (*Grazuleviciene et al., 2009*), the Norwegian Mother, Father and Child Cohort Study (MoBa)(*Magnus et al., 2016*) and the RHEA Mother Child Cohort study in Crete, Greece (*Chatzi et al., 2017*). All participants in the study signed an ethical consent and the study was approved by the ethical committees of each study area (*Maitre et al., 2018*).

In the present study, we selected a total of 832 children of European ancestry that had both DNA methylation and gene expression data. Ancestry was determined with cohort-specific self-reported questionnaires.

### Biological samples

DNA was obtained from buffy coats collected in EDTA tubes at mean age 8.1 years old. Briefly, DNA was extracted using the Chemagen kit (Perkin Elmer), in batches by cohort. DNA concentration was determined in a NanoDrop 1,000 UV-Vis Spectrophotometer (Thermo Fisher Scientific) and with Quant-iT PicoGreen dsDNA Assay Kit (Life Technologies).

RNA was extracted from whole blood samples collected in Tempus tubes (Applied Biosystems) using the MagMAX for Stabilized Blood Tubes RNA Isolation Kit (Thermo Fisher Scientific), in batches by cohort. The quality of RNA was evaluated with a 2100 Bioanalyzer (Agilent) and the concentration with a NanoDrop 1000 UV-Vis Spectrophotometer (Thermo Fisher Scientific). Samples classified as good RNA quality had an RNA Integrity Number (RIN) > 5, a similar RNA integrity pattern at visual inspection, and a concentration >10 ng/μl. Mean values for the RIN, concentration (ng/ul) and Nanodrop 260/230 ratio were: 7.05, 109.07, and 2.15, respectively.

### DNA methylation assessment

DNA methylation was assessed with the Infinium HumanMethylation450K BeadChip (Illumina), following manufacturer's protocol at the National Spanish Genotyping Centre (CEGEN), Spain. Briefly, 700 ng of DNA were bisulfite-converted using the EZ 96-DNA methylation kit following the manufacturer's standard protocol, and DNA methylation measured using the Infinium protocol. A HapMap sample was included in each plate. In addition, 24 HELIX inter-plate duplicates were included. Samples were randomized considering cohort, sex, and panel. Paired samples from the panel study (samples from the same subject collected at different time points) were processed in the same array. Two samples were repeated due to their overall low quality.

DNA methylation data was pre-processed using *minfi* R package (RRID:SCR_012830) (*Aryee et al., 2014*). We increased the stringency of the detection p-value threshold to <1e-16, and probes not reaching a 98% call rate were excluded (*Lehne et al., 2015*). Two samples were filtered due to overall quality: one had a call rate <98% and the other did not pass quality control parameters of the *MethylAid* R package (RRID:SCR_002659) (*van Iterson et al., 2014*). Then, data was normalized with the functional normalization method with Noob background subtraction and dye-bias correction

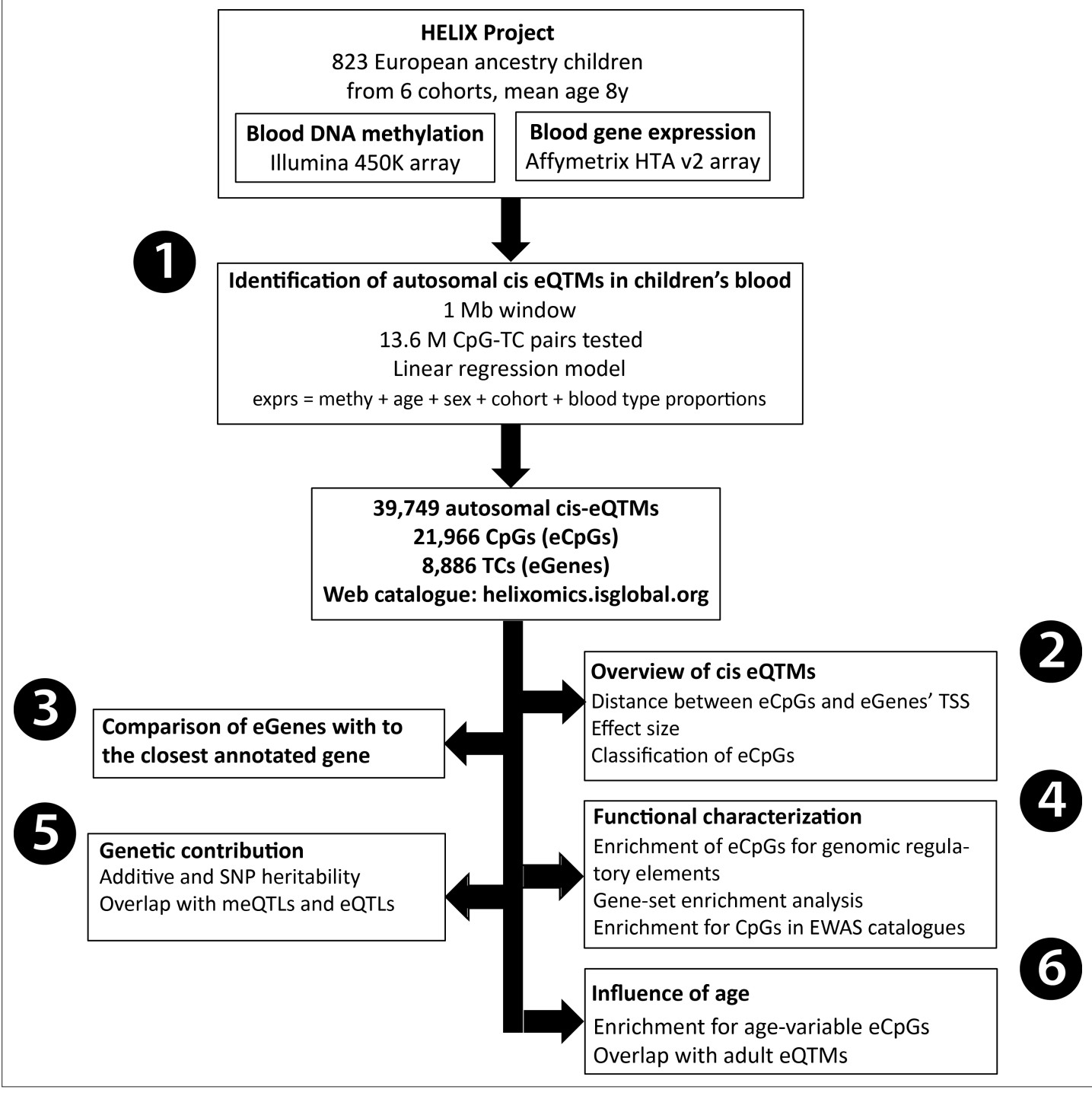

**Figure 1.** Analysis workflow. The figure summarizes the analyses conducted in this study. The first step was (1) the identification of blood autosomal cis eQTMs (1 Mb window centered at the transcription start site, TSS, of the gene) in 823 European ancestry children from the HELIX project, by linear regression models adjusted for age, sex, cohort, and blood cell type proportions. All the associations are reported in the web catalogue (http://www.helixomics.isglobal.org/) and in Dryad (doi:10.5061/dryad.fxpnvx0t0). Then, (2) we explored the distance from the eCpG (CpG involved in an eQTM) to eGene's TSS (gene involved in an eQTM), the effect size of the association, and classified eCpGs in different types. Next, (3) we evaluated the proportion of eGenes potentially inferred through annotation of eCpGs to the closest gene. Finally, (4) we functionally characterized eCpGs and eGenes; (5) assessed the contribution of genetic variants; and (6) evaluated the influence of age.

The online version of this article includes the following figure supplement(s) for figure 1:

**Figure supplement 1.** Distribution of enes and CpGs in all CpG-Gene pairs.

*Figure 1 continued on next page*

**Figure supplement 3.** DNA methylation range by CpG type.

**Figure supplement 5.** Genes call rate distribution by Gene type.

(*Fortin et al., 2014b*). Then, we checked sex consistency using the *shinyMethyl* R package (*Fortin et al., 2014a*), genetic consistency of technical duplicates, biological duplicates (panel study), and other samples making use of the genotype probes included in the Infinium HumanMethylation450K BeadChip and the genome-wide genotyping data, when available. In total four samples were excluded, two with discordant sex and two with discordant genotypes. Batch effect (slide) was corrected using the *ComBat* R package (RRID:SCR_010974) (*Johnson et al., 2007*). Duplicated samples, one of the samples from the panel study and HapMap samples were removed as well as control probes, probes in sexual chromosomes, probes designed to detect Single Nucleotide Polymorphisms (SNPs) and probes to measure methylation levels at non-CpG sites, giving a final number of 386,518 probes.

CpG annotation was conducted with the *IlluminaHumanMethylation450kanno.ilmn-12.hg19* R package (*Hansen, 2016*). Briefly, this package annotates CpGs to proximal promoter (200 bp upstream the TSS - TSS200), distant promoter (from 200 to 1500 bp upstream the TSS - TSS1500), 5'UTR, first exon, gene body, and 3'UTR regions. CpGs farther than 1,500 bp from the TSS were not annotated to any gene. Relative position to CpG islands (island, shelve, shore, and open sea) was also provided by the same R package.

Annotation of CpGs to 15 chromatin states was retrieved from the Roadmap Epigenomics Project web portal (RRID:SCR_008924) (https://egg2.wustl.edu/roadmap/web_portal/). Each CpG in the array was annotated to one or several chromatin states by taking a state as present in that locus if it was described in at least 1 of the 27 blood-related cell types.

## Gene expression assessment

Gene expression, including coding and non-coding transcripts, was assessed with the Human Transcriptome Array 2.0 ST arrays (HTA 2.0) (Affymetrix) at the University of Santiago de Compostela (USC), Spain. Amplified and biotinylated sense-strand DNA targets were generated from total RNA. Affymetrix HTA 2.0 arrays were hybridized according to Affymetrix recommendations using the Manual Target preparation for GeneChip Whole Transcript (WT) expression arrays and the labeling and hybridization kits. In each round, several batches of 24–48 samples were processed. Samples were randomized within each batch considering sex and cohort. Paired samples from the panel study were processed in the same batch. Two different types of control RNA samples (HeLa or FirstChoice Human Brain Reference RNA) were included in each batch, but they were hybridized only in the first batches. Raw data were extracted with the AGCC software (Affymetrix) and stored into CEL files. Ten samples failed during the laboratory process (seven did not have enough cRNA or ss-cDNA, 2 had low fluorescence, and one presented an artifact in the CEL file).

Data was normalized with the GCCN (SST-RMA) algorithm at the gene level. Annotation of transcript clusters (TCs) was done with the ExpressionConsole software using the HTA-2.0 Transcript Cluster Annotations Release na36 annotation file from Affymetrix. After normalization, several quality control checks were performed and four samples with discordant sex and two with low call rates were excluded (*Buckberry et al., 2014*). One of the samples from the panel study was also eliminated for this analysis. Control probes and probes in sexual chromosomes or probes without chromosome information were excluded. Probes with a DABG (Detected Above Background) p-value < 0.05 were considered to have an expression level different from the background, and they were defined as detected. Probes with a call rate <1% were excluded from the analysis. The final dataset consisted of 58,254 TCs.

Gene expression values were $\log_2$ transformed and batch effect controlled by residualizing the effect of surrogate variables calculated with the sva method (RRID:SCR_012836) (*Leek and Storey, 2007*) while protecting for main variables in the study (cohort, age, sex, and blood cellular composition).

## Blood cellular composition

Main blood cell type proportions (CD4+ and CD8+ T cells, natural killer cells, monocytes, eosinophils, neutrophils, and B-cells) were estimated using the Houseman algorithm (*Houseman et al., 2012*) and the Reinius reference panel (*Reinius et al., 2012*) from raw methylation data.

## Genome-wide genotyping

Genome-wide genotyping was performed using the Infinium Global Screening Array (GSA) MD version 1 (Illumina), which contains 692,367 variants, at the Human Genomics Facility (HuGe-F), Erasmus MC, The Netherlands. Genotype calling was done using the GenTrain2.0 algorithm based on a custom cluster file implemented in the GenomeStudio software (RRID:SCR_010973). Annotation was done with the GSAMD-24v1-0_20011747_A4 manifest. Samples were genotyped in two rounds, and 10 duplicates were included which confirmed high inter-round consistency.

Quality control was performed with the PLINK program (RRID:SCR_001757) following standard recommendations (*Chang et al., 2015*; *Purcell et al., 2007*). We applied the following sample quality controls: sample call rate <97% (N filtered = 43), sex concordance (N = 8), heterozygosity based on >4 SD (N = 0), relatedness with PI_HAT > 0.185 (N = 10, including potential DNA contamination), duplicates (N = 19). Then, we used the *peddy* tool (RRID:SCR_017287) to predict ancestry from GWAS data (*Pedersen and Quinlan, 2017*). We contrasted ancestry predicted from GWAS with ancestry recorded in the questionnaires. Twelve samples were excluded due to discordances between the two variables. Overall, 93 (6.7%) samples, including the duplicates, were filtered out. The variant quality control included the following steps: variant call rate <95% (N filtered = 4,046), non-canonical pseudo-autosomal regions (PAR) (N = 47), minor allele frequency (MAF) <1% (N = 178,017), Hardy-Weinberg equilibrium (HWE) p-value < 1e-06 (N = 913). Some other SNPs were filtered out during the matching between data and reference panel before imputation (N = 14,436).

Imputation of the GWAS data was performed with the Imputation Michigan server (RRID:SCR_017579) (*Das et al., 2016*) using the Haplotype Reference Consortium (HRC) cosmopolitan panel, Version r1.1 2016 (*McCarthy et al., 2016*). Before imputation, PLINK GWAS data was converted into VCF format and variants were aligned with the reference genome. The phasing of the haplotypes was done with Eagle v2.4 (RRID:SCR_017262) (*Loh et al., 2016*) and the imputation with minimac4 (RRID:SCR_009292) (*Fuchsberger et al., 2015*), both implemented in the code of the Imputation Michigan server. In total, we retrieved 40,405,505 variants after imputation. Then, we applied the following QC criteria to the imputed dataset: imputation accuracy (R2) >0.9, MAF >1%, HWE p-value > 1e-06; and genotype probabilities were converted to genotypes using the best guest approach. The final post-imputation quality-controlled dataset consisted of 1,304 samples and 6,143,757 variants (PLINK format, Genome build: GRCh37/hg19,+ strand).

## Identification of autosomal cis eQTMs in children's blood

To test associations between DNA methylation levels and gene expression levels in cis (cis eQTMs), we paired each Gene to CpGs closer than 500 kb from its TSS, either upstream or downstream. For each Gene, the TSS was defined based on HTA-2.0 annotation, using the start position for transcripts in the + strand, and the end position for transcripts in the - strand. CpGs position was obtained from Illumina 450 K array annotation. Only CpGs in autosomal chromosomes (from chromosome 1–22) were tested. In the main analysis, we fitted for each CpG-Gene pair a linear regression model between gene expression and methylation levels adjusted for age, sex, cohort, and blood cell type composition. A second model was run without adjusting for blood cellular composition and it is only reported on the online web catalog, but not discussed in this manuscript. Although some of the unique associations of the unadjusted model might be real, others might be confounded by the large methylation and expression changes among blood cell types.

To ensure that CpGs paired to a higher number of enes do not have higher chances of being part of an eQTM, multiple-testing was controlled at the CpG level, following a procedure previously applied in the Genotype-Tissue Expression (GTEx) project (*Gamazon et al., 2018*). Briefly, our statistic used to test the hypothesis that a pair CpG-Gene is significantly associated is based on considering the lowest p-value observed for a given CpG and all its paired Gene (e.g. those in the 1 Mb window centered at the TSS). As we do not know the distribution of this statistic under the null hypothesis, we used a permutation test. We generated 100 permuted gene expression datasets and ran our previous linear

regression models obtaining 100 permuted p-values for each CpG-Gene pair. Then, for each CpG, we selected among all CpG-Gene pairs the minimum p-value in each permutation and fitted a beta distribution that is the distribution we obtain when dealing with extreme values (e.g. minimum) (*Dudbridge and Gusnanto, 2008*). Next, for each CpG, we took the minimum p-value observed in the real data and used the beta distribution to compute the probability of observing a lower p-value. We defined this probability as the empirical p-value of the CpG. Then, we considered as significant those CpGs with empirical p-values to be significant at 5% false discovery rate (FDR) using Benjamini-Hochberg method. Finally, we applied a last step to identify all significant CpG-Gene pairs for all eCpGs. To do so, we defined a genome-wide empirical p-value threshold as the empirical p-value of the eCpG closest to the 5% FDR threshold. We used this empirical p-value to calculate a nominal p-value threshold for each eCpG, based on the beta distribution obtained from the minimum permuted p-values. This nominal p-value threshold was defined as the value for which the inverse cumulative distribution of the beta distribution was equal to the empirical p-value. Then, for each eCpG, we considered as significant all eCpG-Gene variants with a p-value smaller than nominal p-value.

## Characterization of the child blood autosomal cis eQTM catalogue

Wilcoxon tests were run to compare continuous variables (e.g. methylation range, CpG probe reliability, etc.) vs. categorical variables (e.g. low, medium, and high categories of methylation levels, eCpGs vs non eCpGs, etc.). We run a linear regression model to test the association between the effect size and the distance between the CpG and the Gene's TSS. For this test, we compared the absolute value of the effect size vs $\log_{10}$ of absolute value of the distance.

### Enrichment of eCpGs for regulatory elements

Enrichment of eQTMs for regulatory elements were tested using Chi-square tests with non eQTMs as reference, unless otherwise stated. Results with a p-value < 0.05 were considered statistically significant. Annotation of eQTMs to regulatory elements (gene relative positions, CpG island relative positions, and blood ROADMAP chromatin states) is described in the section 'DNA methylation assessment'. Enrichment for CpGs classified in three groups based on their median methylation levels (low: 0.0–0.3; medium: > 0.3–0.7; and high: > 0.7–1.0) was tested similarly.

### Enrichment of eCpGs for CpGs associated with phenotypic traits and exposures

We also explored the enrichment of eQTMs for phenotypic traits and/or environmental exposures reported in the EWAS catalog (*Battram et al., 2021*) and the EWAS Atlas (*Li et al., 2019*). We used version 03-07-2019 of the EWAS catalog and selected those studies conducted in whole or peripheral blood of European ancestry individuals. We downloaded EWAS Atlas data on 27-11-2019 and selected those studies performed in whole blood or peripheral blood of European ancestry individuals or with unreported ancestry. Enrichment was tested as indicated above.

### Enrichment of eCpGs for age-variable CpGs

We used results from the MeDALL and the Epidelta projects to test whether eQTMs were enriched for CpGs variable from birth to childhood and adolescence. For MeDALL, we downloaded data from supplementary material of the following manuscript that assesses changes from 0 to 4y and from 4y to 8y (*Xu et al., 2017*). For Epidelta, we downloaded the full catalogue (version 2020-07-17) from their website (http://epidelta.mrcieu.ac.uk/). In Epidelta, we considered a CpG as age-variable if its p-value from model one that assesses linear changes from 0 to 17 years (variable M1.change.p) was <1e-7 (Bonferroni threshold as suggested in the study). Variable CpGs were classified as increased methylation if their change estimate (variable M1.change.estimate) was >0, and as decreased methylation, otherwise. Enrichment was tested as indicated above.

### Enrichment of eGenes for Gene Ontology - Biological Processes (GO-BP)

We also tested whether eGenes were enriched for specific GO-BP terms using the *topGO* R package (RRID:SCR_014798)(*Alexa and Rahnenfuhrer, 2010*) and using the genes annotated by Affymetrix in our dataset as background (58,254 enes annotated to 23,054 Gene Symbols). We applied the

*weight01* algorithm, which considers GO-BP terms hierarchy for p-values computation. GO-BP terms with q-value <0.001 were considered statistically significant.

## Comparison of genes associated with eQTMs versus annotation of eQTMs to the closest gene

We evaluated whether genes associated with eQTMs could be captured through the Illumina annotation, which links CpGs to the closest gene in a maximum distance of 1,500 bp. For this, CpGs were annotated to Gene Symbols using the *IlluminaHumanMethylation450kanno.ilmn-12.hg19* R package (**Hansen, 2016**), while enes were annotated to Gene Symbols using the HTA-2.0 Transcript Cluster Annotations Release na36 annotation file from Affymetrix. Given that CpGs and enes could be annotated to several genes, we considered that a CpG-Gene pair was annotated to the same gene if at least one of the genes annotated to the CpG was present among the genes in the HTA-2.0 array. In total, we identified 327,931 CpG-Gene pairs annotated to the same gene, and thus that could be compared. Then, a Chi-square test was applied to compute whether eQTMs were enriched for these 327,931 comparable CpG-Gene pairs, using as background all 13 M CpG-Gene pairs.

Next, we evaluated whether the relative position of the CpG in the genic region was related to the expression of the paired Gene. To do so, the comparable 327,931 CpG-Gene pairs were expanded to 383,672 entries. Each entry represented a CpG-Gene pair annotated to a unique gene relative position. Thus, for instance, a CpG-Gene pair with the CpG annotated to two relative gene positions of the same gene was included as two entries, each time annotated to a different gene relative position. In this expanded CpG-Gene pair set, Chi-square tests were run to test the enrichment of eQTMs for gene relative positions, using the 383,672 entries as background.

## Evaluation of the genetic contribution on child blood autosomal cis eQTMs

We used two approaches to evaluate the influence of genetic effects in child blood autosomal cis eQTMs. First, we analyzed heritability estimates of CpGs computed by Van Dongen and colleagues (**van Dongen et al., 2016**). Total additive and SNP-heritabilities were compared between eCpGs and non eCpGs, using a Wilcoxon test. We also run linear regressions between heritability measures (outcome) and eCpGs classified according to the number of eGenes they were associated with.

Second, we tested whether eCpGs were more likely regulated by SNPs than non eCpGs (i.e. whether they were enriched for meQTL). In order to define meQTLs in HELIX, we selected 9.9 M cis and trans meQTLs with a p-value < 1e-7 in the ARIES dataset consisting of data from children of 7 years old (**Gaunt et al., 2016**). Then, we tested whether this subset of 9.9 M SNPs were also meQTLs in HELIX by running meQTL analyses using *MatrixEQTL* R package (**Shabalin, 2012**), adjusting for cohort, sex, age, blood cellular composition and the first 20 principal components (PCs) calculated from genome-wide genetic data. We confirmed 2.8 M meQTLs in HELIX (p-value < 1e-7). Trans meQTLs represented <10% of the 2.8 M meQTLs. Enrichment of eCpGs for meQTLs was computed using a Chi-square test, using non eCpGs as background.

Finally, we tested whether meQTLs were also eQTLs for the eGenes linked to the eCpGs. To this end, we run eQTL analyses (gene expression being the outcome and 2.8 M SNPs the predictors) with *MatrixEQTL* adjusting for cohort, sex, age, blood cellular composition and the first 20 GWAS PCs in HELIX. We considered as significant eQTLs the SNP-Gene pairs with p-value < 1e-7 and with the direction of the effect consistent with the direction of the meQTL and the eQTM.

## Comparison with adult blood eQTM catalogues: GTP and MESA

We compared our list of child blood autosomal cis eQTMs obtained in HELIX with the cis and trans eQTMs described in blood of two adult cohorts: GTP and MESA (**Kennedy et al., 2018**). DNA methylation was assessed with the Infinium HumanMethylation450K BeadChip (Illumina) in the three cohorts. In HELIX, gene expression was assessed with the Human Transcriptome Array 2.0 ST arrays (HTA 2.0) (Affymetrix), and in GTP and MESA with the HumanHT-12 v3.0 and v4.0 Expression BeadChip (Illumina).

For the comparison of eQTMs between adults and children, eGenes in the two studies were annotated to a common gene nomenclature, by using the Gene Symbol annotation provided by the authors form GTP and MESA, and the Gene Symbol provided by the Affymetrix annotation in HELIX.

Some eQTMs involved transcript clusters (HELIX) or gene probes (GTP and MESA) annotated to more than one gene (Gene Symbol); and also different enes (HELIX) or gene probes (GTP and MESA) were annotated to the same Gene Symbol. To handle this issue, we split our comparison in two analyses.

First, we checked whether CpG-gene pairs reported in GTP and MESA were eQTMs (significant CpG-gene pairs) in HELIX. By doing this, the comparison was restricted to cis effects (as HELIX only considered cis effects). When a CpG-gene pair in GTP or MESA mapped to multiple CpG-gene pairs in HELIX, we only considered the CpG-gene pair with the smallest p-value in HELIX. Next, Pearson's correlations between the effect sizes of the different studies were computed.

Second, we explored whether HELIX eQTMs were also present in GTP and/or MESA. When a CpG-gene pair in HELIX mapped to multiple CpG-gene pairs in GTP and/or MESA, we only considered the CpG-gene pair with the smallest p-value in these cohorts. As a result, HELIX eQTMs were classified in age-shared (if present in adults at p-value < 1e-05, in GTP and/or MESA) and children-specific (absent in adult cohorts). For these two subsets of eQTMs, enrichment for ROADMAP chromatin states, methylation measurement error, and distance from the eCpG to the eGene's TSS, was tested as explained above.

## Data and software availability

The raw data used to generate the eQTM catalogue are not publicly available due to privacy restrictions but are available from the corresponding author on request. Catalogue of eQTMs described in this manuscript is publicly available at https://helixomics.isglobal.org/ and at Dryad (doi:10.5061/dryad.fxpnvx0t010.5061/dryad.fxpnvx0t0). Scripts to reproduce the analysis can be found in a public GitHub repository (https://github.com/yocra3/methExprsHELIX/) (*Ruiz-Arenas, 2021*) and as a supplementary file.

**Table 1.** Descriptive of the study population.

BiB: Born in Bradford study (UK). EDEN: Étude des Déterminants pré et postnatals du développement et de la santé de l'Enfant (France). KANC: Kaunus cohort (Lithuania). MoBa: Norwegian Mother, Father and Child Cohort Study (Norway). RHEA: Mother Child Cohort study (Greece). INMA: INfancia y Medio Ambiente cohort (Spain).

| Variable | BiB | EDEN | KANC | MoBa | RHEA | INMA | All |
|---|---|---|---|---|---|---|---|
| N (%) | 80 (9.7%) | 80 (9.7%) | 143 (17.4%) | 188 (22.9%) | 154 (18.7%) | 178 (21.6%) | 823 (100%) |
| Female (%) | 36 (45%) | 35 (43.8%) | 64 (44.8%) | 88 (46.8%) | 69 (44.8%) | 80 (44.9%) | 372 (45.2%) |
| Male (%) | 44 (55%) | 45 (56.2%) | 79 (55.2%) | 100 (53.2%) | 85 (55.2%) | 98 (55.1%) | 451 (54.8%) |
| Age, in years (IQR) | 6.65 (6.44–6.84) | 10.76 (10.37–11.22) | 6.40 (6.12–6.88) | 8.53 (8.17–8.83) | 6.45 (6.36–6.62) | 8.84 (8.44–9.21) | 8.06 (6.49–8.86) |
| Natural Killer cells (IQR) | 0.01 (0.00–0.03) | 0.02 (0.00–0.04) | 0.04 (0.01–0.07) | 0.02 (0.00–0.07) | 0.01 (0.00–0.03) | 0.03 (0.01–0.05) | 0.02 (0.00–0.05) |
| B-cell (IQR) | 0.12 (0.11–0.15) | 0.09 (0.07–0.11) | 0.11 (0.09–0.13) | 0.11 (0.09–0.14) | 0.14 (0.11–0.16) | 0.10 (0.08–0.13) | 0.11 (0.09–0.14) |
| CD4+ T cell (IQR) | 0.21 (0.18–0.25) | 0.16 (0.14–0.20) | 0.17 (0.14–0.21) | 0.21 (0.17–0.25) | 0.20 (0.16–0.26) | 0.17 (0.14–0.21) | 0.19 (0.15–0.23) |
| CD8+ T cell (IQR) | 0.13 (0.11–0.17) | 0.11 (0.08–0.13) | 0.13 (0.10–0.16) | 0.14 (0.11–0.17) | 0.14 (0.11–0.16) | 0.12 (0.09–0.14) | 0.13 (0.10–0.16) |
| Monocytes (IQR) | 0.09 (0.07–0.10) | 0.09 (0.07–0.11) | 0.08 (0.06–0.09) | 0.08 (0.07–0.10) | 0.09 (0.07–0.10) | 0.09 (0.07–0.11) | 0.08 (0.07–0.10) |
| Granulocytes (IQR) | 0.41 (0.35–0.47) | 0.52 (0.47–0.56) | 0.46 (0.40–0.53) | 0.41 (0.32–0.48) | 0.41 (0.34–0.48) | 0.48 (0.42–0.55) | 0.44 (0.37–0.52) |

Continuous variables are expressed as mean and interquartile range (IQR).

## Results

### Study population and molecular data

The study includes 823 children of European ancestry from the HELIX project with available blood DNA methylation and gene expression data. These children, enrolled in six cohorts, were aged between 6 and 11 years and the number of males and females was balanced (*Table 1*).

After quality control, our dataset consists of 386,518 CpGs and 58,254 transcript clusters (TCs) in autosomal chromosomes (from 1 to 22). TCs are defined as groups of one or more probes covering a region of the genome, reflecting all the exonic transcription evidence known for the region, and corresponding to a known or putative gene. Thus, we will refer TCs to enes indistinctively. According to Affymetrix annotation, 23,054 of the enes encoded a protein. To detect cis effects, we paired each ene to all CpGs closer than 0.5 Mb from its transcription start site (TSS), either upstream or downstream (1 Mb window centered at the TSS). In total, we obtained 13.6 M CpG-Gene pairs, where each CpG was paired to a median of 30 enes; and each ene was paired to a median of 162 CpGs (*Figure 1—figure supplement 1*).

### Identification of autosomal cis eQTMs in children's blood

We tested the association between DNA methylation and gene expression levels in the 13.6 M autosomal CpG-Gene pairs through linear regressions adjusting for sex, age, cohort, and cellular composition. After correcting for multiple testing (see Material and Methods), we identified 39,749 statistically significant autosomal cis eQTMs in children's blood (0.29% of total CpG-Gene pairs). These eQTMs comprised 21,966 unique CpGs (5.7% of total CpGs) and 8,886 unique enes (15.3% of total enes), of which 6288 were annotated as coding genes. For simplicity, we will refer to them as eQTMs (statistically significant associations of CpG-Gene pairs), eCpGs (CpGs involved in eQTMs), and eGenes (enes involved in eQTMs). 23,355 eQTMs (58.8% of total) showed inverse associations, meaning that higher DNA methylation was associated with lower gene expression. In eQTMs, each eGene was associated with a median of 2 eCpGs, while each eCpG was associated with a median of 1 eGene (*Figure 1—figure supplement 2*). eCpGs presented higher methylation variability in the population (*Figure 1—figure supplement 3*), and had higher intraclass correlation coefficients, a proxy of low technical error (*Sugden et al., 2020*; *Figure 1—figure supplement 4*). Indeed, 13,278 eCpGs (60.4% of total) were measured with probes which had an intraclass correlation coefficient (ICC) > 0.4, which is indicative of reliable measurements. Moreover, eGenes had higher call rates (*Figure 1—figure supplement 5*).

The complete catalogue of eQTMs can be downloaded from https://helixomics.isglobal.org/ and from Dryad (doi:10.5061/dryad.fxpnvx0t0).

### Overview of autosomal cis eQTMs in children's blood

#### Distance from the eCpG to the eGene's TSS and effect size

eCpGs tended to be close to the TSS of the targeted eGenes, being this distance <250 Kb for 87.9% of all eQTMs (*Figure 2A*). Globally, the median distance between an eCpG and the TSS of its associated eGene was 1.1 kb (IQR = –33 kb; 65 kb), being eCpGs closer to the TSS in inverse eQTMs than in positive. The observed downstream shift could be explained because we chose the most upstream TSS for each ene according to the Affymetrix HTAv2 annotation. A similar shift was observed for expression quantitative trait loci, eQTLs, (i.e. single-nucleotide polymorphisms, SNPs, associated with gene expression) in the Genotype-Tissue Expression (GTEx) project (*Gamazon et al., 2018*).

We report the effect size of eQTMs as the $\log_2$ fold change (FC) of gene expression per 0.1 points increase in methylation (or 10 percentile increase). In absolute terms, the median effect size was 0.12, being the minimum 0.002 and the maximum 16.0, with 96.3% of the eQTMs with an effect size < 0.5. A median effect size of 0.12 means that a change of 0.1 points in methylation levels was associated with around a 9% increase/decrease of gene expression. We observed an inverse linear association between the eCpG-eGene's TSS distance and the effect size (p-value = 7.75e-9, *Figure 2B*); while we did not observe significant differences in effect size due to the relative orientation of the eCpG (upstream or downstream) with respect to the eGene's TSS (p-value = 0.68).

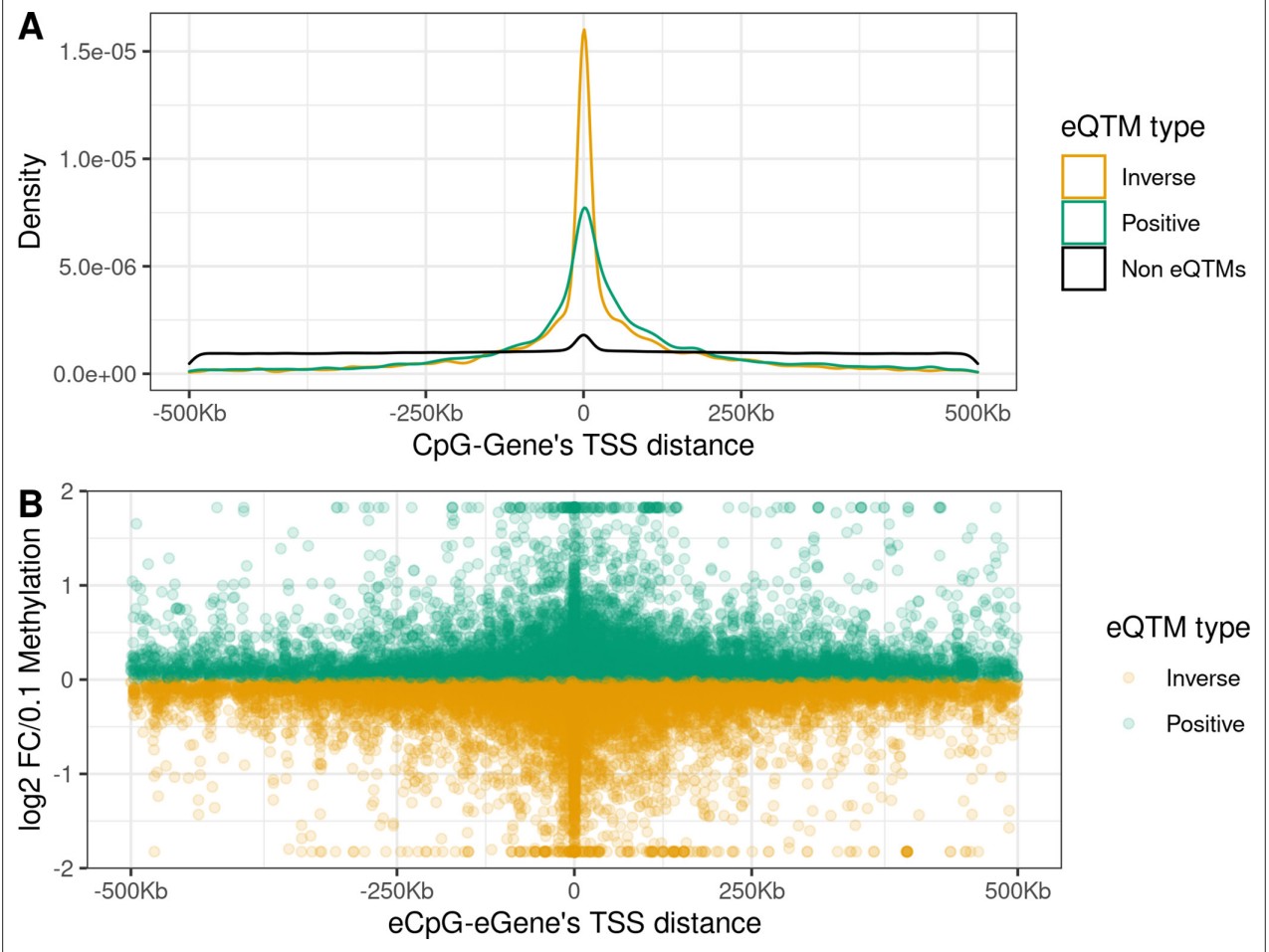

**Figure 2.** Distance between CpG and ene's TSS and effect size in child blood autosomal cis eQTMs. (**A**) Distribution of the distance between CpG and ene's TSS by eQTM type. CpG-Gene pairs were classified in non eQTMs (black); inverse eQTMs (yellow); and positive eQTMs (green). The x-axis represents the distance between the CpG and the ene's TSS (kb). Non eQTMs median distance: –0.013 kb (interquartile range - IQR = –237; 236). Positive eQTMs median distance: –4.9 kb (IQR = –38; 79). Inverse eQTMs median distance: –0.7 kb (IQR = –29; 54). (**B**) Effect size versus eCpG-Gene's TSS distance in eQTMs. The x-axis represents the distance between the eCpG and the eGene's TSS (kb). The y-axis represents the effect size as the log2 fold change in gene expression produced by a 0.1 increase in DNA methylation (or 10 percentile increase). To improve visualization, a 99% winsorization has been applied to log2 fold change values: values more extreme than 99% percentile have been changed for the 99% quantile value. eQTMs are classified in inverse (yellow) and positive (green). Each eQTM is represented by one dot. The darker the color, the more dots overlapping, and so the higher the number of eQTMs with the same effect size and eCpG-eGene's TSS distance.

The online version of this article includes the following figure supplement(s) for figure 2:

**Figure supplement 1.** Enrichment of eCpGs for gene relative positions.

**Table 2.** Classification of eCpGs by type.
Percentages refer to the total number of eCpGs.

|       | Inverse (N, %) | Positive (N, %) | Bivalent (N, %)   | Total (N, %)    |
|-------|----------------|-----------------|-------------------|-----------------|
| Mono  | 8,084 (36.8%)  | 5,681 (25.9%)   | 0, by definition  | 13,765 (62.7%)  |
| Multi | 3,738 (17.0%)  | 2,400 (10.9%)   | 2,063 (9.4%)      | 8,201 (37.3%)   |
| Total | 11,822 (53.8%) | 8,081 (36.8%)   | 2,063 (9.4%)      | 21,966 (100%)   |

## Classification of eCpGs

As shown in *Table 2*, we classified eCpGs into five types, by following two criteria: (1) the number of eGenes affected, distinguishing between mono eCpGs (associated with a unique eGene), and multi eCpGs (associated with ≥ 2 eGenes); and (2) the direction of the effect, distinguishing between inverse, positive and bivalent eCpGs (with inverse effects on some eGenes and positive effects on others). Mono inverse eCpGs were the most abundant type (36.8%) (*Table 2*). CpGs not associated with the expression of any Gene were named as non eCpGs. We used these categories in the subsequent analyses.

## Comparison of eGenes with the closest annotated gene

A standard approach to interpret EWAS findings is to assume that a CpG regulates the expression of proximal genes. These genes are usually identified through the Illumina 450 K annotation (Hansen, n.d.), which annotates a CpG to a gene when the CpG maps into the gene body, untranslated, or promoter region defined as < 1500 bp upstream the TSS. We evaluated to which extent the Illumina 450 K annotation captured the eQTMs identified in our catalogue.

First, we observed that CpG-Gene pairs where CpG and Gene were annotated to the same Gene Symbol were more likely eQTMs than CpG-Gene pairs annotated to different Gene Symbols or without gene annotation (OR = 11.90, p-value < 2e-16). Next, we assessed whether the gene annotated to the eCpG with the Illumina 450 K annotation was coincident with the eGene found in our analysis. To answer this, we selected 14,797 eCpGs (67.4% of total eCpGs) annotated to Gene Symbols also present in the Affymetrix array, and thus comparable. In 7,808 out of these 14,797 eCpGs, the eCpG was associated with the expression of an eGene which was coincident with at least one of the Gene Symbols the eCpG was annotated to (52.8% of eCpGs with comparable gene annotation, 35.5% of all eCpGs).

Finally, we explored whether the relative gene position of a CpG determines its association with gene expression. We selected the 327,931 CpG-Gene pairs with the CpG and Gene annotated to the same Gene Symbol. Within this subset, eCpGs were enriched for CpGs in 5'UTRs and gene body positions, while depleted for CpGs in proximal promoters and 3'UTRs (*Figure 2—figure supplement 1*). Interestingly, we observed that inverse and positive eCpGs were enriched for CpGs located in different gene regions: inverse for CpGs in distal promoters (TSS1500) and 5'UTRs; positive for CpGs in gene bodies.

Overall, only around half of the eGenes targeted by the eQTMs could be identified by the Illumina 450 K annotation. We also found that eCpGs were enriched for TSS1500, 5'UTRs, and gene body positions.

## Functional characterization of autosomal cis eQTMs in children's blood

### Enrichment of eCpGs for genomic regulatory elements

We characterized eCpGs by evaluating their enrichment for diverse regulatory elements, including CpG island relative positions and 15 chromatin states retrieved from 27 blood cell types from the ROADMAP Epigenomics project (*Roadmap Epigenomics Consortium et al., 2015*). First, we found that eCpGs were depleted for CpG islands, while mostly enriched for CpG island shores, but also for shelves and open sea (*Figure 3A*). We did not observe relevant differences between inverse and positive eCpGs.

Second, we assessed whether eCpGs were enriched for ROADMAP blood chromatin states (*Roadmap Epigenomics Consortium et al., 2015*; *Figure 3B*). eCpGs were enriched for several active states, such as enhancers or active transcription regions. Nonetheless, we observed some discrepancies between eCpGs subtypes: only inverse eCpGs were enriched for proximal promoter states while only positive eCpGs were depleted for transcription at 5' and 3' (TxFlnk). In inactive chromatin states, both positive and inverse eCpGs were enriched for bivalent regulatory states (BivReg), while only positive eCpGs were enriched for repressed and weak repressed Polycomb regions (ReprPC, ReprPCWk) and quiescent regions (Quies).

Third, we also analyzed whether eCpGs had different methylation levels. We found that eCpGs were enriched for CpGs with medium ( > 0.3–0.7) methylation levels and depleted for CpGs with low (0–0.3) or high ( > 0.7–1) methylation levels (*Figure 3C*).

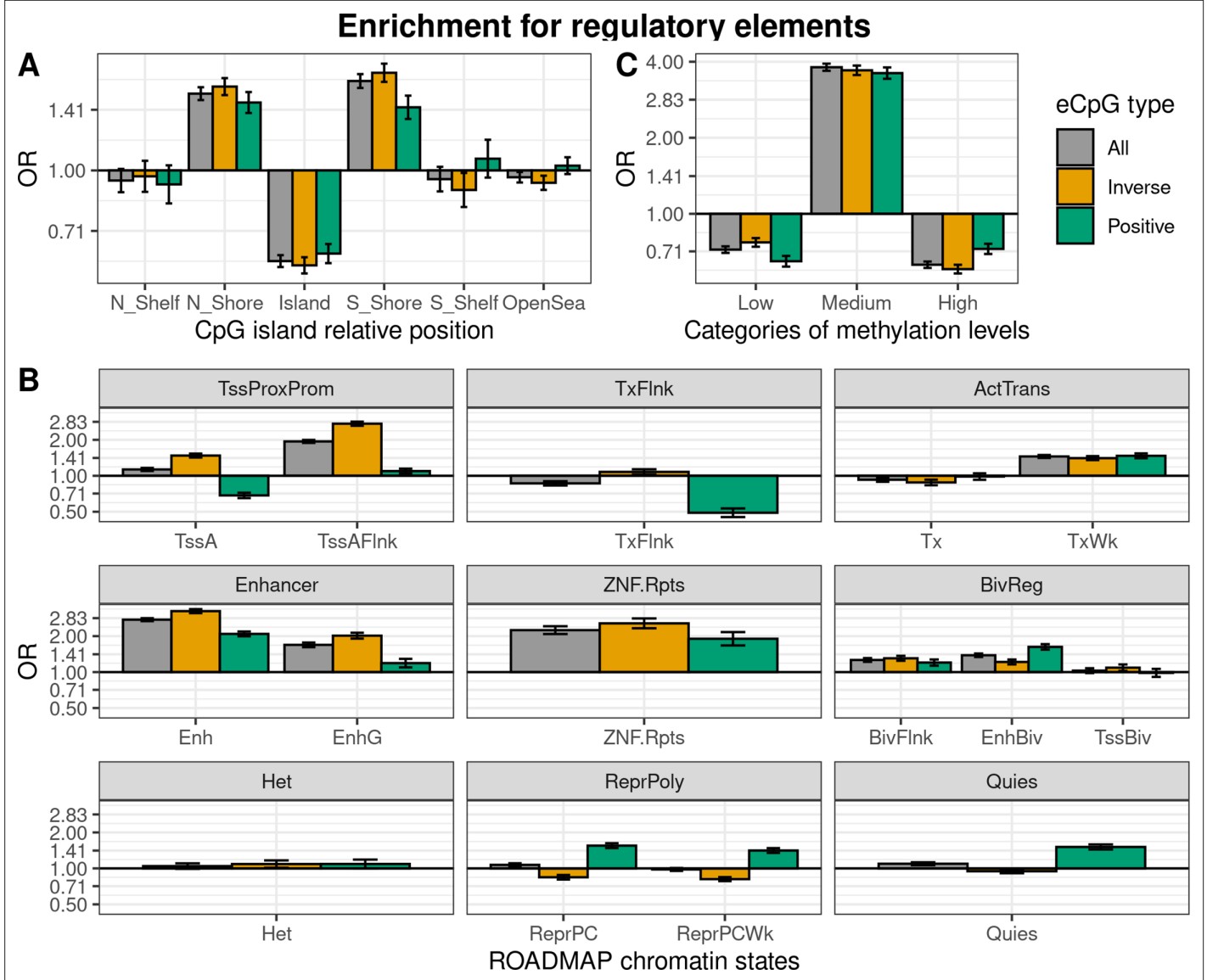

**Figure 3.** Enrichment of cis autosomal eCpGs in children's blood for different regulatory elements. eCpGs were classified in all (grey), inverse (yellow), and positive (green). The y-axis represents the odds ratio (OR) of the enrichment. In all cases, the enrichment was computed against non eCpGs. (**A**) Enrichment for CpG island relative positions: CpG island, N- and S-shore, N- and S-shelf, and open sea. (**B**) Enrichment for ROADMAP blood chromatin states (**_Roadmap Epigenomics Consortium et al., 2015_**): active TSS (TssA); flanking active TSS (TssAFlnk); transcription at 5' and 3' (TxFlnk); transcription region (Tx); weak transcription region (TxWk); enhancer (Enh); genic enhancer (EnhG); zinc finger genes and repeats (ZNF.Rpts); flanking bivalent region (BivFlnx); bivalent enhancer (EnhBiv); bivalent TSS (TssBiv); heterochromatin (Het); repressed Polycomb (ReprPC); weak repressed Polycomb (ReprPCWk); and quiescent region (Quies). Chromatin states can be grouped in active transcription start site proximal promoter states (TssProxProm), active transcribed states (ActTrans), enhancers (Enhancers), bivalent regulatory states (BivReg), and repressed Polycomb states (ReprPoly). (**C**) Enrichment for categories of CpGs with different median methylation levels: low (0–0.3), medium (0.3–0.7), and high (0.7–1) (**_Huse et al., 2015_**).

The online version of this article includes the following figure supplement(s) for figure 3:

**Figure supplement 1.** Enrichment of eCpGs with reliable measurement for different regulatory elements.

**Figure supplement 2.** Enrichment of autosomal cis eCpGs in children's blood for CpGs reported to be associated with phenotypic traits and/or environmental exposures.

Finally, we wondered whether these enrichments could be affected by the bias introduced by methylation measurement error; thus, we repeated all the enrichment analyses only considering 75,836 CpGs measured with reliable probes (ICC > 0.4) (**_Sugden et al., 2020_**; **_Figure 3—figure supplement 1_**). After this filtering, the enrichments for CpG island relative positions and for categories of CpGs

according to their methylation levels changed substantially: eCpGs passed from being depleted to being enriched for CpG island positions (*Figure 3—figure supplement 1A*), and from being enriched for CpGs with medium methylation levels to being enriched for CpGs with low methylation levels (*Figure 3—figure supplement 1C*). On the contrary, the magnitudes of enrichments for most of the active chromatin states were increased (*Figure 3—figure supplement 1B*); while enrichments of positive eCpGs for inactive states (ReprPoly and Quies) were reverted. Overall, selecting reliable CpG probes reduced the differences between inverse and positive eCpGs and resulted in enrichments for active chromatin states and depletions for inactive states.

## Gene-set enrichment analysis

To identify which biological functions were regulated by our list of eQTMs, we ran gene-set enrichment analyses using the list of eGenes. 5503 out of the 8886 unique Gene Symbols annotated to eGenes were present in Gene Ontology - Biological Processes (GO-BP), leading to 52 enriched terms (q-value < 0.001) (*Supplementary file 1A*). As expected from the tissue analyzed, 50% of the terms were related to immune responses (N = 26), followed by terms associated with cellular (N = 16) and metabolic (N = 10) processes. Among immune terms, 9 of them were part of innate immunity, 9 of adaptive response, and eight were related to general/other immune pathways. Most enriched GO-BP terms were also found when running the enrichment with the list of eGenes derived from eQTMs measured with reliable CpG probes (ICC > 0.4) (*Supplementary file 1A*).

## Enrichment for CpGs reported in the EWAS catalogues

We assessed whether eCpGs were enriched for CpGs previously related to phenotypic traits and/or environmental exposures. To this end, we retrieved CpGs from EWAS performed in blood of European ancestry subjects: 143,384 CpGs from the EWAS catalog (*Battram et al., 2021*), and 54,599 CpGs from the EWAS Atlas (*Li et al., 2019*). We found that eCpGs were enriched for CpGs in these EWAS databases in comparison to non eCpGs. Although we observed larger odds ratios (ORs) for CpGs listed in the EWAS Atlas than for CpGs in the EWAS Catalog (*Figure 3—figure supplement 2A*), this difference disappeared after removing CpGs with less reliable measurements (ICC < 0.4) (*Figure 3—figure supplement 2B*).

## Genetic contribution to autosomal cis eQTMs in children's blood

### Additive and SNP heritability of eQTMs

We hypothesized that genetic variation might regulate DNA methylation and gene expression in some of the autosomal cis eQTMs in children's blood. To test this, we used two measures of genetic influence: (1) heritability of blood DNA methylation levels for each CpG, calculated from twin designs (total additive heritability) and from genetic relationship matrices (SNP heritability), as reported by Van Dongen and colleagues (*van Dongen et al., 2016*) and (2) methylation quantitative trait loci (meQTLs, SNPs associated with DNA methylation levels) identified in the ARIES dataset (*Gaunt et al., 2016*).

First, we found that eCpGs had higher total additive and SNP heritabilities than non eCpGs (median difference of 0.31 and 0.11, respectively, p-value < 2e-16 for both). Moreover, total additive and SNP heritabilities were higher for eCpGs associated with a larger number of eGenes (increase of 0.025 and 0.026 points per eGene, respectively, with a p-value < 2e-16 for both) (*Figure 4A and B*). After removing CpG probes with unreliable measurements (ICC < 0.4), differences in median total additive heritability between eCpGs and non eCpGs were still present, but smaller (0.15, p-value < 2e-16); whereas differences in SNP heritabilities were maintained (0.11, p-value < 2e-16) (*Figure 4—figure supplement 1*).

### Overlap with methylation and expression quantitative trait loci (meQTLs and eQTLs)

Second, we studied whether eCpGs were enriched for meQTLs, either in cis or trans. We analyzed 1,078,466 meQTLs identified in blood samples of 7-year-old children in the ARIES dataset and replicated in HELIX (see Material and Methods). These meQTLs affected the methylation of 36,671 CpGs through a total of 2,820,145 SNP-CpG pairs. 10,187 eCpGs (27.8% total eCpGs) presented at least one meQTL, being eCpGs enriched in CpGs associated with genetic variants (OR: 11.06, p-value < 2e-6). In addition, among CpGs with meQTLs, eCpGs were associated with a higher number of

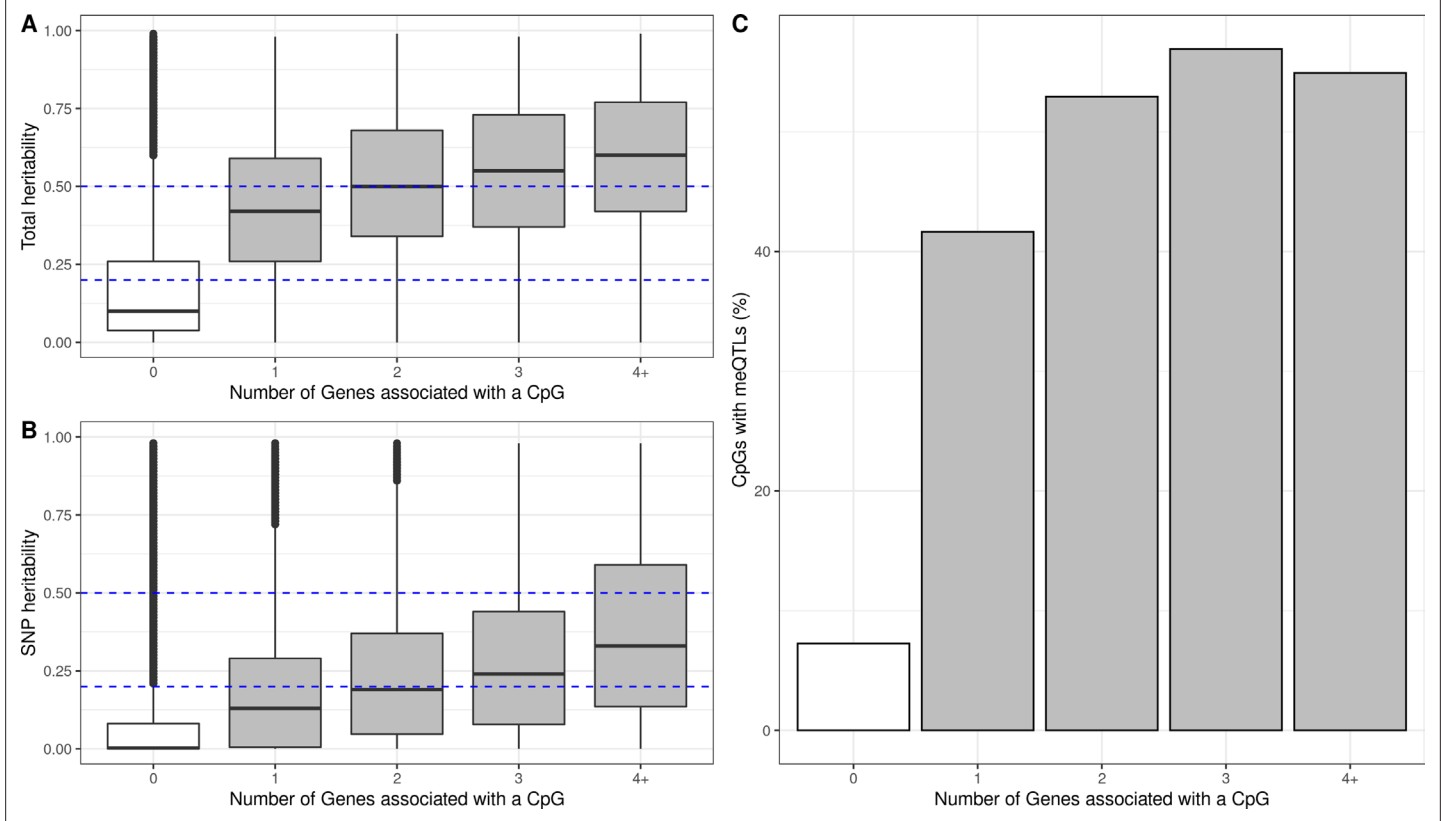

**Figure 4.** Genetic contribution to autosomal cis eQTMs in children's blood. CpGs were grouped by the number of Genes they were associated with, where 0 means that a CpG was not associated with any Gene (non eCpG). (**A**) Total additive heritability and (**B**) SNP heritability as inferred by Van Dongen and colleagues (***van Dongen et al., 2016***). The y-axis represents heritability and the x-axis each group of CpGs associated with a given number of Genes. (**C**) Proportion of CpGs having a meQTL (methylation quantitative trait locus), by each group of CpGs associated with a given number of Genes.

The online version of this article includes the following figure supplement(s) for figure 4:

**Figure supplement 1.** Heritability of methylation levels in CpGs with reliable measurements.

**Figure supplement 2.** Proportion of CpGs having a meQTL (methylation quantitative trait loci) among CpGs with reliable measurements.

**Figure supplement 3.** Probe reliability in autosomal cis eCpGs according to association with genetic variants.

**Figure supplement 4.** Example of a trio of SNP-CpG-Gene.

meQTLs (median: 74, IQR: 27; 162) than non eCpGs (median: 32, IQR = 10; 77). Finally, eCpGs associated with a higher number of eGenes are more likely to be associated with at least one meQTL (***Figure 4C***). After removing CpG probes with unreliable measurements (ICC < 0.4), we observed the same trends, although the enrichment of eCpGs for CpGs with at least one meQTL was reduced (OR = 3.5, p-value < 2.2e-16) (***Figure 4—figure supplement 2***). Finally, we observed that eCpGs with at least one meQTL were measured with higher reliability (higher ICC) than eCpGs without any meQTL (***Figure 4—figure supplement 3***).

We, then, determined whether SNPs associated with eCpGs were also associated with the corresponding eGenes. After multiple-testing correction, we identified 1,368,613 SNP-CpG-Gene trios with consistent direction of effect, and 12,799 with inconsistent direction. These formers comprised 16,055 unique eQTMs (40.4% of significant eQTMs); 8503 unique eCpGs (38.7% of total eCpGs); and 4098 unique eGenes (46.1% of total eGenes), of which 3154 were coding (50.2% of total coding eGenes). In these trios, eGenes were associated with a median of 2 eCpGs (IQR = 1; 5) and 67 SNPs (IQR = 21; 149); whereas eCpGs were associated with a median of 1 eGene (IQR = 1; 2) and 53 SNPs (IQR = 17; 124). One example of such a SNP-CpG-Gene trio is formed by rs11585123-cg15580684-TC01000080.hg.1 (*AJAP1*), in chromosome 10 (***Figure 4—figure supplement 4***).

Next, we run gene-set enrichment analyses with the 2,746 eGenes involved in these trios. We identified 35 significant GO-BP terms (q-value < 0.001). Of these, 14 were related to immunity (six innate, four adaptive immunity, and four general/other); 11 to cellular processes; and 10 to metabolic processes (*Supplementary file 1A*). In comparison to all eGenes, eGenes under genetic control had a reduction in the number of GO-BP terms involving immune and cellular functions (*Supplementary file 1B*).

Overall, we found that a substantial part of the eQTMs seems to be under genetic control, and the SNPs associated with DNA methylation levels of eCpGs were also associated with gene expression levels of eGenes.

## Influence of age on autosomal cis eQTMs in children's blood

### Enrichment for CpG whose methylation change with age

To understand the association between changes in methylation and gene expression throughout life, first we evaluated whether eCpGs were enriched for CpGs whose methylation levels change from birth to childhood/adolescence according to literature. To this end, we retrieved the CpGs that vary with age from two databases (*Mulder et al., 2021*; *Xu et al., 2017*): the MeDALL project (*Xu et al., 2017*) which described 14,150 CpGs whose methylation change between 0 and 8 years (9,647 with increased and 4,503 with decreased methylation); and the Epidelta project (*Mulder et al., 2021*), which describes 244,283 CpGs whose methylation change between 0 and 17 years (168,314 with increased and 75,969 with decreased methylation) from. Of note, 90% of the CpGs identified in the MeDALL project were also reported in the Epidelta. We found that eCpGs were enriched for CpGs whose methylation change in both MeDALL and Epidelta databases, but more markedly for CpGs reported in MeDALL (*Figure 5A*). In both databases, positive and inverse eCpGs showed stronger ORs for CpGs with increased and decreased methylation levels over age, respectively. After excluding CpG probes with unreliable measurements (ICC < 0.4), MeDALL enrichments were reduced to the magnitude of Epidelta enrichments, while the differences between positive and inverse eCpGs were more evident (*Figure 5—figure supplement 1*).

### Overlap with autosomal eQTMs in adult blood

We evaluated whether autosomal cis eQTMs in children's blood were consistent in adult populations. For this, we used data from the study of autosomal cis and trans eQTMs in adults' blood based on two cohorts: (1) GTP, whole blood and 333 samples; and (2) MESA, monocytes and 1,202 samples, by Kennedy and colleagues (*Kennedy et al., 2018*). The catalogue contains the summary statistics of all autosomal cis ( < 50 kb from the TSS) and trans (otherwise) CpG-gene pairs at p-value < 1e-5, although only CpG-gene associations at p-value < 1e-11 were considered significant eQTMs in their study. To compare their findings with ours, we mapped Genes and gene probes to Gene Symbols and compared CpG-gene pairs (see Materials and methods, *Supplementary file 1C*).

We observed that 57.9% and 35.3% of eQTMs with p-value < 1e-5 in GTP and MESA were also eQTMs in HELIX, thus age-shared eQTMs (*Figure 5B*). More than 90% of age-shared eQTMs have the same direction in GTP/MESA than in HELIX (*Supplementary file 1D*). In addition, effect sizes in GTP/MESA were correlated with effects sizes in HELIX (*Supplementary file 1D*).

Only 5471 (13.8%) of the eQTMs identified in HELIX children were reported in adult GTP or MESA catalogues at p-value < 1e-5 (*Figure 5C*). We explored whether eQTMs identified both in HELIX children and in adults (age-shared eQTMs) had different characteristics compared to eQTMs only found in children (child-specific eQTMs). Age-shared eQTMs involved 4364 eCpGs and 1689 eGenes, whereas children-specific eQTMs involved 19,584 eCpGs and 8429 eGenes. Age-shared eCpGs had higher reliability (higher ICC) (*Figure 5—figure supplement 2*) and tended to be closer to the TSS than child-specific eCpGs (*Figure 5—figure supplement 3*). The enrichment for ROADMAP blood chromatin states (*Roadmap Epigenomics Consortium et al., 2015*) of age-shared and child-specific eCpGs in comparison to non eCpGs was quite similar (*Figure 5—figure supplement 4*). Nonetheless, age-shared eCpGs showed higher ORs of enrichment for proximal promoters. Both types of eCpGs were enriched for meQTLs compared to non eCpGs, with the OR being stronger for age-shared eCpGs (OR = 20.7) than for child-specific eCpGs (OR = 10.3).

Overall, we found that eQTMs were enriched for CpGs whose methylation levels changed from birth to adolescence. The overlap between child and adult eQTMs was small: only 13.8% of HELIX

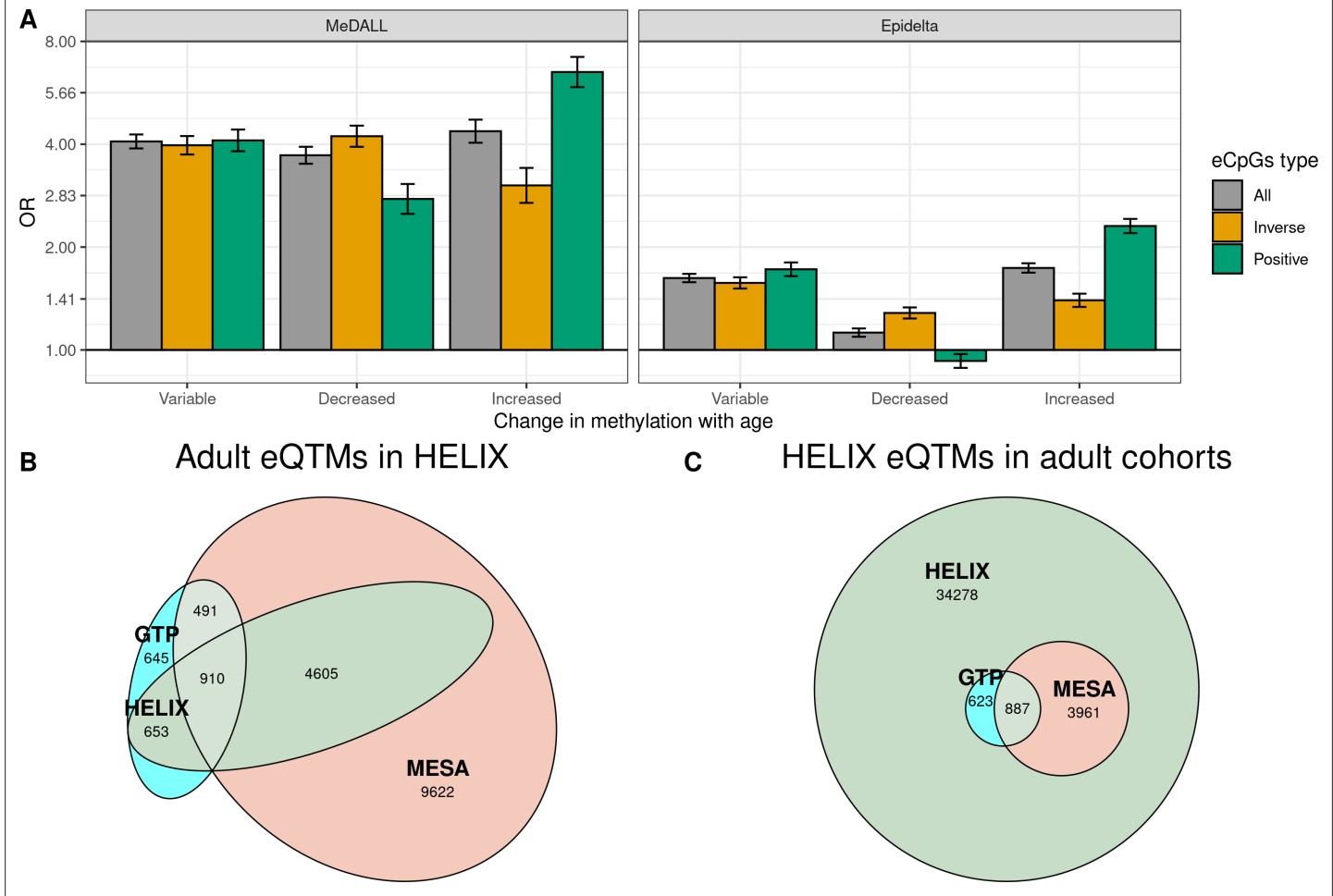

**Figure 5.** Influence of age on autosomal cis eQTMs in children's blood. (**A**) Enrichment of eCpGs for CpGs whose methylation levels change with age, in comparison to non eCpGs. eCpGs were classified in all (grey); inverse (yellow); and positive (green). CpGs whose methylation change with age were retrieved from the MeDALL project (from birth to childhood **Xu et al., 2017**) and from the Epidelta project (from birth to adolescence **Mulder et al., 2021**). They were classified in variable (CpGs with methylation levels that change with age); decreased (CpGs with methylation levels that decrease with age); and increased (CpGs with methylation levels that increase with age). The y-axis represents the odds ratio (OR) of the enrichment. (**B**) Overlap between autosomal cis/trans eQTMs identified in adults (GTP: whole blood; MESA: monocytes) (**Kennedy et al., 2018**) with cis eQTMs identified in children (HELIX: whole blood). All CpG-gene pairs reported at *P*-value < 1e-5 in GTP or MESA that could be compared with pairs in HELIX are shown. (**C**) Overlap between blood autosomal cis eQTMs identified in HELIX children with cis/trans eQTMs identified in adults (GTP: whole blood; MESA: monocytes) (**Kennedy et al., 2018**). All CpG-gene pairs in HELIX that could be compared with pairs in GTP or MESA are shown. Note: The comparison has been split into two plots because one eGene in HELIX can be mapped to different expression probes in GTP and MESA, and vice-versa. Only comparable CpG-Gene pairs are shown (see Materials and methods).

The online version of this article includes the following figure supplement(s) for figure 5:

**Figure supplement 1.** Enrichment of eCpGs with reliable measurements for CpGs with age-variable methylation levels.

**Figure supplement 2.** Probe reliability in eCpGs according to overlap with adult eQTMs.

**Figure supplement 3.** Distribution of the distance between CpG-Gene's TSS by eQTM type.

**Figure supplement 4.** Enrichment of age-shared and child-specific eCpGs for blood ROADMAP blood chromatin states.

eQTMs had also been described in adults. Age-shared eCpGs tended to be proximal to the TSS, enriched for promoter chromatin states, and with stronger signals of genetic regulation.

## Discussion

In this work, we present a blood autosomal cis eQTM catalogue in children. We identified 39,749 eQTMs, representing 21,966 unique eCpGs and 8886 unique eGenes (6288 of which were coding).

23,355 eQTMs (58.8% of all eQTMs) showed inverse associations. A substantial fraction was influenced by genetic variation, and the overlap with eQTMs reported in adults was small.

The characteristics of the autosomal cis eQTMs in children's blood were highly consistent with patterns previously described in other studies. Most of the eCpGs tended to be proximal to the eGene's TSS (*Kennedy et al., 2018*; *Taylor et al., 2019*). The magnitude of the effect seemed to be proportional to the distance between the eCpG and the eGene's TSS, but this association was weak. Although higher DNA methylation is assumed to lead to lower expression, we found that around 40% of eQTMs were positively associated with gene expression. This percentage is in line with previous results from different tissues (*Gutierrez-Arcelus et al., 2015*; *Gutierrez-Arcelus et al., 2013*; *Küpers et al., 2019*). Inverse and positive eCpGs tended to be localized in enhancers and other active regulatory regions and not in CpG islands, a pattern that was also previously reported (*Gutierrez-Arcelus et al., 2015*; *Küpers et al., 2019*). Despite these common locations, inverse eCpGs were specifically found around active TSSs (including the distal promoter and the 5'UTR), while positive eCpGs were localized in gene body regions. These results highlight the importance of the genomic context to infer the direction of the association between DNA methylation and gene expression (*Kennedy et al., 2018*). We want to point out that the causal relationship between DNA methylation and gene expression cannot be definitely inferred from our study. Indeed, there is some evidence suggesting that DNA methylation could be a consequence of gene expression, as opposed to the often assumed concept that regulation of gene expression is mediated by DNA methylation (*Gutierrez-Arcelus et al., 2013*; *Jones, 2012*; *Kim et al., 2020*). eQTMs can be influenced by genetic variation (*Lu et al., 2019*). In HELIX, eCpGs linked to the expression of several eGenes had higher heritabilities and were associated with a higher number of meQTLs than non eCpGs. This could suggest that eCpGs that regulate the expression of several genes, the so-called master regulators, are more prone to be themselves regulated by genetic variation. We, then, searched for SNPs simultaneously associated with DNA methylation (meQTLs) and gene expression (eQTLs) in our data. We identified 1.3 M SNP-CpG-Gene trios with consistent direction of the effect. Interestingly, the number of GO-BP terms related to immune and cellular functions was reduced for eGenes under genetic control, in comparison to all eGenes; on the contrary, the number of GO-BP terms involving metabolic processes was maintained. This may suggest that the influence of environmental factors is more relevant for immune pathways, while genetic factors might be more determinant in regulating metabolic processes in blood cells. Given the non-negligible effect of genetics in eQTMs, we would advise studying the effect of genetic variants on the association between environmental factors or phenotypic traits and DNA methylation.

In order to know how eQTMs behave along life-course, we compared blood autosomal cis eQTMs identified in HELIX children with cis and trans eQTMs reported by Kennedy and colleagues in whole blood and monocytes from adult populations (*Kennedy et al., 2018*). We found that only 13.8% of the autosomal eQTMs in children's blood were also reported in adults. Similarly, a modest proportion of adult blood eQTMs was present in children (58% from GTP and 35% from MESA). This small overlap between adult and child eQTMs has different explanations: methodological issues, such as gene expression platforms with low overlap; statistical methods and statistical power; cohort-specific environmental exposures; and cellular composition. Unsurprisingly, HELIX and MESA presented the highest divergence, as HELIX assessed eQTMs in whole blood and MESA in monocytes. Despite the effect of these methodological and confounding factors, it is known that DNA methylation and gene expression change with age (*Melé et al., 2015*; *Mulder et al., 2021*; *Xu et al., 2017*); consequently, we could expect only partial overlap between adult and child eQTMs. The short list of age-shared eCpGs tended to encompass CpGs located in promoters and regulated by genetic variants. Moreover, the overall location of eQTMs in regulatory elements was similar between adults and children (*Gutierrez-Arcelus et al., 2015*; *Küpers et al., 2019*). This could represent a specific characteristic of eQTMs that are persistent over time. An alternative explanation is that this kind of eQTMs (genetically regulated and close to the TSS) are easier to be detected and shared among any two studies because they show stronger effects. Finally, we observed that HELIX eQTMs usually involved CpGs whose methylation varied between birth and childhood/adolescence, and they tended to activate rather than inactivate transcription over this period. Also, they were enriched for CpGs found to be related to environmental factors and phenotypic traits in the EWAS Atlas and EWAS Catalog.

As previously described (*Sugden et al., 2020*), CpG probes have different measurement error and thus different reliability and reproducibility. Consequently, CpGs measured with less error have

more chances of being found associated with traits and thus reported in EWAS catalogues. In HELIX, we found that CpG probe ICC was higher for these different cases: for eCpGs, in comparison to non eCpGs; for age-shared eCpGs, in comparison to children-specific eCpGs; and for eCpGs with meQTLs, in comparison to eCpGs without meQTLs. In this line, enrichments of eCpGs for CpGs listed in the EWAS Atlas or in the MeDALL project were markedly attenuated when only considering CpGs measured with good reliability. Moreover, CpG probe reliability is dependent on DNA methylation level and variance (highly unmethylated or highly methylated CpGs, which tend to have low variances, are measured with more error); and genomic regulatory elements are characterized by particular methylation levels. Therefore, this biased the enrichments for regulatory elements. For instance, after considering only reliable probes, the distribution of eQTMs in CpG island relative positions changed completely (*Figure 3—figure supplement 1*). Moreover, the enrichments for active chromatin states were amplified and differences between inverse and positive eCpGs attenuated.

Our study of autosomal cis eQTMs in children's blood has several strengths compared to previous eQTM studies. First, we report all CpG-gene pairs we tested in our analysis, as opposed to existing blood eQTM catalogues which only reported pairs passing a given p-value threshold (*Bonder et al., 2017*; *Kennedy et al., 2018*). Reporting all pairs tested allows replication and meta-analyses, reducing publication bias. Second, we report which eQTMs are influenced by genetic variation, and researchers can take this into account when exploring the relationship between methylation and expression in their data. Finally, as others (*Wu et al., 2018*), we describe that only around half of the CpG-Gene relationships are captured through annotation to the closest gene. Therefore, our eQTM catalogue becomes an essential and powerful tool to help researchers interpret their EWAS, with a particular focus on childhood.

The catalogue also has some limitations. First, it only covers a fraction of all CpG-Gene pairs, as both the methylation and gene expression arrays have limited resolution. Nonetheless, the catalogue will be useful for most researchers as the methylation array is widely used, and the gene expression array covers almost all the coding genes. Second, the catalogue does not include sex chromosomes which require more complex analyses to address X-inactivation and sex-specific effects that will be addressed in future studies. Third, due to statistical power limitations, only cis effects were tested. Despite that, we observed that eCpGs tended to be close to the gene they regulate, so the catalogue is expected to cover most of the CpG-Gene associations. Fourth, effect sizes should be considered with caution as the association between DNA methylation and gene expression might be non-linear, and the effect of outlier values was not systematically explored (*Johnson et al., 2017*). Fifth, models were adjusted for blood cell type composition and, while this has allowed us to control for major differences in methylation and gene expression among blood cell types, it might also have resulted in over-adjustment in some CpG-Gene pairs. Moreover, the analysis of bulk data might have limited the identification of eQTMs specific to a subset of blood cell types, the identification of which would need more sophisticated statistical and/or experimental methods. Finally, we acknowledge that the catalogue will be useful for biological interpretation of EWAS if it is true that DNA methylation is not a mere mark of cell memory to past exposures (without transcriptional consequences or with time-limited ones) (*Tsai et al., 2018*).

In summary, besides characterizing child blood autosomal cis eQTMs and reporting how they are affected by genetics and age, we provide a unique public resource: a catalogue with 13.6 M CpG-gene pairs and of 1.3 M SNP-CpG-gene trios (https://helixomics.isglobal.org/). This information will improve the biological interpretation of EWAS findings.

## Acknowledgements

The authors acknowledge the contribution of all the HELIX children and their families. Funding The study has received funding from the European Community's Seventh Framework Programme (FP7/2007-206) under grant agreement no 308,333 (HELIX project); the H2020-EU.3.1.2. - Preventing Disease Programme under grant agreement no 874,583 (ATHLETE project); from the European Union's Horizon 2020 research and innovation programme under grant agreement no 733,206 (LIFECYCLE project), and from the European Joint Programming Initiative "A Healthy Diet for a Healthy Life" (JPI HDHL and Instituto de Salud Carlos III) under the grant agreement no AC18/00006 (NutriPROGRAM project). The genotyping was supported by the projects PI17/01225 and PI17/01935, funded by the Instituto de Salud Carlos III and co-funded by European Union (ERDF, "A way to make Europe") and

the Centro Nacional de Genotipado-CEGEN (PRB2-ISCIII). BiB received core infrastructure funding from the Wellcome Trust (WT101597MA) and a joint grant from the UK Medical Research Council (MRC) and Economic and Social Science Research Council (ESRC) (MR/N024397/1). INMA data collections were supported by grants from the Instituto de Salud Carlos III, CIBERESP, and the Generalitat de Catalunya-CIRIT. KANC was funded by the grant of the Lithuanian Agency for Science Innovation and Technology (6-04-2014_31 V-66). The Norwegian Mother, Father and Child Cohort Study is supported by the Norwegian Ministry of Health and Care Services and the Ministry of Education and Research. The Rhea project was financially supported by European projects (EU FP6-2003-Food-3-NewGeneris, EU FP6. STREP Hiwate, EU FP7 ENV.2007.1.2.2.2. Project No 211,250 Escape, EU FP7-2008-ENV-1.2.1.4 Envirogenomarkers, EU FP7-HEALTH-2009- single stage CHICOS, EU FP7 ENV.2008.1.2.1.6. Proposal No 226,285 ENRIECO, EU- FP7- HEALTH-2012 Proposal No 308333 HELIX), and the Greek Ministry of Health (Program of Prevention of obesity and neurodevelopmental disorders in preschool children, in Heraklion district, Crete, Greece: 2011–2014; "Rhea Plus": Primary Prevention Program of Environmental Risk Factors for Reproductive Health, and Child Health: 2012–15). We acknowledge support from the Spanish Ministry of Science and Innovation through the "Centro de Excelencia Severo Ochoa 2019–2023" Program (CEX2018-000806-S), and support from the Generalitat de Catalunya through the CERCA Program. MV-U and CR-A were supported by a FI fellowship from the Catalan Government (FI-DGR 2015 and #016FI_B 00272). MC received funding from Instituto Carlos III (Ministry of Economy and Competitiveness) (CD12/00563 and MS16/00128).

# Additional information

## Competing interests

The authors declare that no competing interests exist.

## Funding

| Funder | Grant reference number | Author |
| --- | --- | --- |
| Wellcome Trust | WT101597MA | John Wright |
| European Commission | 308333 | Martine Vrijheid |
| Instituto de Salud Carlos III | AC18/00006 | Martine Vrijheid |
| Medical Research Council | MR/N024397/1 | John Wright |
| Economic and Social Research Council | MR/N024397/1 | John Wright |
| CIBERESP | | Martine Vrijheid |
| Generalitat de Catalunya | 016FI_B 00272 | Carlos Ruiz-Arenas |
| Lithuanian Agency for Science Innovation and Technology | 6-04-2014_31V-66 | Regina Grazuleviciene |
| Ministry of Health and Care Services | | Kristine Bjerve Gutzkow |
| Greek Ministry of Health | FP6-2003-Food-3-NewGeneris | Leda Chatzi |
| Ministerio de Ciencia, Innovación y Universidades | CD12/00563 | Maribel Casas |
| Instituto de Salud Carlos III | PI17/01225 | Mariona Bustamante |
| Instituto de Salud Carlos III | PI17/01935 | Mariona Bustamante |

The funders had no role in study design, data collection and interpretation, or the decision to submit the work for publication.

## Author contributions
Carlos Ruiz-Arenas, Conceptualization, Formal analysis, Methodology, Writing – original draft, Supervision; Carles Hernandez-Ferrer, Conceptualization, Methodology, Supervision; Marta Vives-Usano, Writing – review and editing; Sergi Marí, Formal analysis; Ines Quintela, Dan Mason, Solène Cadiou, Maribel Casas, Sandra Andrusaityte, Kristine Bjerve Gutzkow, Marina Vafeiadi, Leda Chatzi, Ángel Carracedo, Xavier Estivill, Eulàlia Marti, Writing – review and editing, Supervision; John Wright, Johanna Lepeule, Regina Grazuleviciene, Data curation, Supervision; Geòrgia Escaramís, Methodology, Supervision; Martine Vrijheid, Data curation, Funding acquisition, Supervision; Juan R González, Methodology, Resources, Supervision; Mariona Bustamante, Conceptualization, Resources, Writing – original draft, Supervision

## Author ORCIDs
Carlos Ruiz-Arenas (iD) http://orcid.org/0000-0002-6014-3498
Sergi Marí (iD) http://orcid.org/0000-0003-2023-195X
Johanna Lepeule (iD) http://orcid.org/0000-0001-8907-197X
Mariona Bustamante (iD) http://orcid.org/0000-0003-0127-2860

## Ethics
All participants in the study signed an ethical consent and the study was approved by the ethical committees of each study area.

## Decision letter and Author response
Decision letter https://doi.org/10.7554/eLife.65310.sa1
Author response https://doi.org/10.7554/eLife.65310.sa2

---

# Additional files

## Supplementary files
• Supplementary file 1. Supplementary tables. (A) Enrichment of eGenes for GO-BP terms. (B) Summary of GO-BP terms of all eGenes, eGenes of eQTMs measured with reliable CpG probes (ICC > 0.4), and eGenes under genetic control (with meQTLs/eQTLs). (C) Summary of autosomal cis eQTMs in adults' blood. (D) Correlation and concordance of effect sizes between eQTMs in HELIX and in GTP and MESA.

• Transparent reporting form

• Source code 1. Code used to run the analyses and generate the tables and figures.

## Data availability
The eQTM catalogue, summarized data generated in the analysis and which is described in the manuscript, is publicly available at https://helixomics.isglobal.org/ and at Dryad (https://doi.org/10.5061/dryad.fxpnvx0t0). Scripts to reproduce the analysis can be found in a public github repository (https://github.com/yocra3/methExprsHELIX/; copy archived at swh:1:rev:4eb5f17b3e2551364d4aa2e-98be32ff222af8e0d) and as a supplementary file. The individual level data used to generate the eQTM catalogue are not publicly available for several reasons. First, HELIX participants were not explicitly informed about this in the informed consent. Second, there are some studies that suggest that DNA methylation data has enough information to identify participants. Third, each HELIX cohort follows different internal regulations in regards to public access of the data. Nonetheless, individual level data can still be shared with external researchers after signature of a data transfer agreement (DTA). More information to initiate the request process can be can be found at https://www.projecthelix.eu/index.php/es/data-inventory.

The following dataset was generated:

| Author(s) | Year | Dataset title | Dataset URL | Database and Identifier |
|---|---|---|---|---|
| Ruiz-Arenas C, Bustamante M | 2021 | eQTM catalogue in children's blood | https://doi.org/10.5061/dryad.fxpnvx0t0 | Dryad Digital Repository, 10.5061/dryad.fxpnvx0t0 |

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
