## [Editor Report]

The study investigates, for the first time, gene regulation by DNA methylation across the genome in children. The results will be useful for better interpreting the many completed and ongoing studies of the effects of environmental and disease on DNA methylation in children. Prior to this study, investigators had to make do with inaccurate information derived from adult studies.

---

## [Decision Letter]

**Decision letter after peer review:**

Thank you for submitting your article "Identification of blood autosomal cis-expression quantitative trait methylation (cis-eQTMs) in children" for consideration by *eLife*. Your article has been reviewed by 3 peer reviewers, including Matthew Suderman as the Reviewing Editor and Reviewer #3, and the evaluation has been overseen by Patricia Wittkopp as the Senior Editor.

Essential revisions:

1. Gene-CpG associations should only be reported for the model adjusting for cellular composition. By failing to account for cellular composition, the associations identified by this model are difficult to interpret as they may be driven by the extensive gene expression and DNA methylation differences between cell types, true biological dependencies between CpG sites and genes within one or more cell types, or a combination of both.

The authors incorrectly claim that associations identified in the unadjusted model but not in the adjusted model are cell-type-specific associations (e.g. line 417). While possible, it is more likely that the associations are entirely due to differences between cell types (which tend to be large) with little or no association within any specific cell type (which tend to be much smaller). The supporting F-statistic analysis actually provides support for the former possibility, not the latter as claimed in the text. This argument should be removed or revised.

2. Methods used for calculating empirical p-values/significance thresholds need to be clarified and possibly corrected. The authors claim to have applied the approach used by Bonder et al. and first described by Westra et al. (Nat Genet 2013), but the brief description of the approach in the paper is not consistent with Westra et al.

3. Several of the reported enrichments could be driven by variation in probe quality across the DNA methylation arrays. For example, enrichment for eQTM CpG sites among those that change with age could simply be due to the fact that age and eQTM effects are more likely to be observed for CpG sites with high quality probes than low quality probes. This possibility should at least be acknowledged for specific enrichment results likely to be affected.

4. The manuscript in several places does not summarize results well. Long lists of items already provided in tables and figures should be replaced with more readable summaries. Details of less important findings should be moved to the supplementary and replaced with summarizing text.

*Reviewer #1 (Recommendations for the authors):*

Major weakness 1 – With regard to the models used to find eQTMs:

This issue is easily resolved, as the authors already have the results from the fully adjusted model in hand. The authors state that the results of the fully adjusted model are a subset of the unadjusted model, with limited exceptions (line 424). With their cell-type adjusted results being robust, it would be prudent and more widely accepted (and my recommendation) to use the cell-type proportion adjusted model as their main model, that is used in all of the follow-up analyses.

Major weakness 2 – With regard to the authors hypothesis of eQTM associations unique to models unadjusted for cell-type proportion variability across samples:

1) I recommend that the issue of cell-specific vs general CpG-gene associations in unadjusted models should be removed from this manuscript and peer reviewed on its own in a separate manuscript. A new method (or a novel application of a previous method) should not be embedded in an applied paper with little statistical support. If the authors hypothesis is correct and the associations found in an unadjusted model represent cell-type-specific associations – that result would have wide-ranging implications and deserves more focus than it gets in this manuscript.

2) If it is the authors wish to keep this section in this manuscript (which I still do not recommend), it could work as a follow-up analysis with the use of a previously peer reviewed method that is meant specifically to find cell-specific associations. A method that comes to mind is TOAST 1.

1. Li Z, Wu H. TOAST: improving reference-free cell composition estimation by cross-cell type differential analysis. Genome Biol. 2019 Sep 4;20(1):190. PMCID: PMC6727351

Major weakness 3 – p-value adjustment:

If the authors generated an empirical FDR as Bonder and Westra did, I request that they adjust their written methods to make it clear that their methods were the same as those cited. However, if their method is a significant deviation from the methods used by Bonder and Westra, I think the authors should either use an established method for FDR, or at the very least compare their results to a traditional method, as suggested by another reviewer.

*Reviewer #3 (Recommendations for the authors):*

1. Report only results from the cell-type adjusted models.

2. Enrichments prone to probe quality bias should be omitted or flagged in the text.

3. Much of the paper is unnecessarily difficult to read. Many sections contain high levels of detail which could easily be summarized and the details looked up in provided tables or figures (e.g. "CpGs in eQTMs are enriched for blood active chromatin states and medium methylation levels" and "Annotating CpGs to the closest gene only partially captures eQTMs").

4. The Discussion section contains a lot of awkward phrases making it difficult to read.

[Editors' note: further revisions were suggested prior to acceptance, as described below.]

Thank you for submitting your article "Identification of autosomal cis expression quantitative trait methylation (cis eQTMs) in children's blood" for consideration by *eLife*. Your article has been reviewed by 2 peer reviewers, and the evaluation has been overseen by a Reviewing Editor and Kathryn Cheah as the Senior Editor. The reviewers have opted to remain anonymous.

Essential revisions:

1) Please respond to the comments from reviewer 2 below. However, we do *not* require any new analyses (e.g. mediation) or replications to be performed. You also do *not* need to move any text or results to supplementary materials.

2) There was some initial confusion about accessing the data on the website. Perhaps data access information on the website could be made clearer.

3) Reviewer 2 found the "Enrichment for age-variable eQTMs" section difficult to understand. In particular, it could be made clearer that the age-associated CpG sites were identified in previously published studies (MeDALL and Epidelta) and were not the results of your own analyses.

4) The references to "variable methylation" in this section is misleading because it suggests variation that has not been accounted for. However, the variation here is explained by associations with age.

*Reviewer #1 (Recommendations for the authors):*

In this manuscript, "Identification of autosomal cis expression quantitative trait methylation (cis eQTMs) in children's blood", the authors describe the landscape of DNA methylation, in relation to proximal gene expression. In addition to providing a useful resource for researchers conducting EWAS in children's whole blood, the authors do a really great job building on the work that others have done before. I believe the authors have sufficiently responded to reviewer comments.

*Reviewer #2 (Recommendations for the authors):*

The authors replied to reviewers' comments well.

Website is still PW protected.

Suggest defining autosomal in abstract and introduction.

General comments: Try to include all results you made to respond the reviewers' comments in either the main manuscript or supplementary material. Readers may have the same question as the reviewers. For example, include a figure where x-axis is the distance between the CpG site and TC TSS and y-axis is p-value in Supplementary material.

Comments on the manuscript:

Great improvement on writing. Now the manuscript is readable, and now I can see thoroughness of your analysis.

In general, you still have too many results with too many details. Consider moving some of the findings to supplementary. I suggested several results to move to supplementary materials. Move figure and tables that are included in the text to supplementary materials as well.

Major comments:

1. Including results with ICC<0.4 in addition to the main results (without the screening) throughout the manuscript are disturbing. You can choose either all eQTM results or eQTM with ICC<0.4 result throughout the manuscript and move the other result to supplementary material.

2. Line 645-646: "we identified 1,368,613 SNP-CpG-Gene trios with consistent direction of effect, and 12,799 with inconsistent direction – Did you do mediation analysis? If you did not, conduct mediation analysis (such as Baron and Kenny or Sobel) and rewrite this paragraph based on the mediation analysis result. Also make it clear what are consistent direction and inconsistent directions are if you decide this sentence in the manuscript.

3. The section "Enrichment for age-variable eQTMs" is very confusing.

a. Did you conduct EWAS of age in the cohort?

b. Term "variable methylation is confusing". If you did EWAS, then call EWAS methylation.

c. How many methylation CpGs did you test in each cohort?

d. The results from Epidelta data is not very interesting. Move this result to supplementary section.

4. You did replication of eQTMs in adults' cohorts only. I wonder if you can find a children's cohort and try to replicate your eQTMs results in that cohort.

---

## [Author Response]

Essential revisions:1. Gene-CpG associations should only be reported for the model adjusting for cellular composition. By failing to account for cellular composition, the associations identified by this model are difficult to interpret as they may be driven by the extensive gene expression and DNA methylation differences between cell types, true biological dependencies between CpG sites and genes within one or more cell types, or a combination of both.The authors incorrectly claim that associations identified in the unadjusted model but not in the adjusted model are cell-type-specific associations (e.g. line 417). While possible, it is more likely that the associations are entirely due to differences between cell types (which tend to be large) with little or no association within any specific cell type (which tend to be much smaller). The supporting F-statistic analysis actually provides support for the former possibility, not the latter as claimed in the text. This argument should be removed or revised.

Following reviewers’ recommendations, we have reconsidered our initial hypothesis about the role of cellular composition in the association between methylation and gene expression. Although we still think that some of the eQTMs only found in the model unadjusted for cellular composition could represent cell specific effects, we acknowledge that the majority might be confounded by the extensive gene expression and DNA methylation differences between cell types. Also, we recognize that more sophisticated statistical tests should be applied to prove our hypothesis. Because of this we have decided to report the eQTMs of the model adjusted for cellular composition in the main manuscript and keep the results of the model unadjusted for cellular composition only in the online catalogue.

2. Methods used for calculating empirical p-values/significance thresholds need to be clarified and possibly corrected. The authors claim to have applied the approach used by Bonder et al. and first described by Westra et al. (Nat Genet 2013), but the brief description of the approach in the paper is not consistent with Westra et al.

We apologize for this misleading citation. Although Bonder et al. applied a

permutation approach to adjust for multiple-testing, our approach was inspired by the method applied in the GTEx project (GTEX consortium, 2020), using CpGs instead of SNPs. We have improved the description of the method and corrected the citation in the manuscript.

3. Several of the reported enrichments could be driven by variation in probe quality across the DNA methylation arrays. For example, enrichment for eQTM CpG sites among those that change with age could simply be due to the fact that age and eQTM effects are more likely to be observed for CpG sites with high quality probes than low quality probes. This possibility should at least be acknowledged for specific enrichment results likely to be affected.

We agree that some of the reported enrichments could be driven by the intrinsic technical properties of each one of the probes. Because of this, we identified probes with intraclass correlation coefficient (ICC) > 0.4 according to Sudgen and colleagues (Sudgen et al., 2020), and repeated some of the downstream analyses. In particular, we repeated these enrichments:

– Enrichment for different regulatory elements: Figure 3—figure supplement 1;

– Enrichment for CpGs reported to be associated with phenotypic traits and/or environmental exposures: Figure 3—figure supplement 1B;

– Heritability of methylation levels: Figure 4—figure supplement 1 ;

– Proportion of CpGs having meQTLs (methylation quantitative trait loci): Figure 4—figure supplement 2;

– Enrichment for CpGs with age variable methylation levels: Figure 5—figure

supplement 1.

In addition, we also analyzed the distribution of the ICC for different types of CpGs:

– Probe reliability by CpG type (eQTM vs non-eQTM): Figure 1—figure supplement 4;

– Probe reliability by eQTM type (genetically determined eQTM vs not genetically determined eQTM): Figure 4—figure supplement 3;

– Probe reliability by eQTM type (age-shared eQTM vs age-specific eQTM): Figure 5—figure supplement 2.

These analyses are shown as supplemental figures and discussed briefly in the main text.

Finally in the Discussion section we have a specific comment on this issue (page 32 – line 761):

“As previously described (Sugden et al., 2020), CpG probes have different measurement error and thus different reliability and reproducibility. […] Moreover, the enrichments for active chromatin states were amplified and differences between inverse and positive eCpGs attenuated.”

4. The manuscript in several places does not summarize results well. Long lists of items already provided in tables and figures should be replaced with more readable summaries. Details of less important findings should be moved to the supplementary and replaced with summarizing text.

We have simplified the manuscript by keeping only the most relevant findings in the main text, main figures and tables. The remaining results have been moved to Supplemental Tables. Also, since the comparison between the adjusted and unadjusted models for cellular composition has been withdrawn, now the text is easier to read.

Reviewer #1 (Recommendations for the authors):Major weakness 1 – With regard to the models used to find eQTMs:This issue is easily resolved, as the authors already have the results from the fully adjusted model in hand. The authors state that the results of the fully adjusted model are a subset of the unadjusted model, with limited exceptions (line 424). With their cell-type adjusted results being robust, it would be prudent and more widely accepted (and my recommendation) to use the cell-type proportion adjusted model as their main model, that is used in all of the follow-up analyses.

Following reviewers’ recommendations, we have reconsidered our initial hypothesis about the role of cellular composition in the association between methylation and gene expression. Although we still think that some of the eQTMs only found in the model unadjusted for cellular composition could represent cell specific effects, we acknowledge that the majority might be confounded by the extensive gene expression and DNA methylation differences between cell types. Also, we recognize that more sophisticated statistical tests should be applied to prove our hypothesis. Because of this we have decided to report the eQTMs of the model adjusted for cellular composition in the main manuscript and keep the results of the model unadjusted for cellular composition only in the online catalogue.

Major weakness 2 – With regard to the authors hypothesis of eQTM associations unique to models unadjusted for cell-type proportion variability across samples:1) I recommend that the issue of cell-specific vs general CpG-gene associations in unadjusted models should be removed from this manuscript and peer reviewed on its own in a separate manuscript. A new method (or a novel application of a previous method) should not be embedded in an applied paper with little statistical support. If the authors hypothesis is correct and the associations found in an unadjusted model represent cell-type-specific associations – that result would have wide-ranging implications and deserves more focus than it gets in this manuscript.

We agree with the reviewer that additional evidence, with more sophisticated statistical methods and additional data, should be provided to confirm our hypothesis of cell type specific eQTMs. Because of this, as commented in the previous point, now the manuscript focuses on the eQTMs identified in the model adjusted for cellular composition.

2) If it is the authors wish to keep this section in this manuscript (which I still do not recommend), it could work as a follow-up analysis with the use of a previously peer reviewed method that is meant specifically to find cell-specific associations. A method that comes to mind is TOAST 1.1. Li Z, Wu H. TOAST: improving reference-free cell composition estimation by cross-cell type differential analysis. Genome Biol. 2019 Sep 4;20(1):190. PMCID: PMC6727351

We thank the reviewer for his/her suggestion. Nonetheless, and as recommended, we will consider these follow-up analyses to identify cell type specific eQTMs in another study.

Major weakness 3 – p-value adjustment:If the authors generated an empirical FDR as Bonder and Westra did, I request that they adjust their written methods to make it clear that their methods were the same as those cited. However, if their method is a significant deviation from the methods used by Bonder and Westra, I think the authors should either use an established method for FDR, or at the very least compare their results to a traditional method, as suggested by another reviewer.

We apologize for this misleading citation. Although Bonder et al. applied a permutation approach to adjust for multiple testing, our approach was inspired by the method applied in the GTEx project (GTEX consortium, 2020), using CpGs instead of SNPs. The citation has been corrected in the manuscript. Moreover, we have explained in more detail the whole multiple-testing processes in the Material and Methods section.

Reviewer #3 (Recommendations for the authors):1. Report only results from the cell-type adjusted models.

Following reviewers’ recommendations, we have reconsidered our initial hypothesis about the role of cellular composition in the association between methylation and gene expression. Although we still think that some of the eQTMs only found in the model unadjusted for cellular composition could represent cell specific effects, we acknowledge that the majority might be confounded by the extensive gene expression and DNA methylation differences between cell types. Also, we recognize that more sophisticated statistical tests should be applied to prove our hypothesis. Because of this we have decided to report the eQTMs of the model adjusted for cellular composition in the main manuscript and keep the results of the model unadjusted for cellular composition only in the online catalogue.

2. Enrichments prone to probe quality bias should be omitted or flagged in the text.

We agree that some of the reported enrichments could be driven by the intrinsic technical properties of each one of the probes. Because of this, we identified probes with intraclass correlation coefficient (ICC) > 0.4 according to Sudgen and colleagues (Sudgen et al., 2020), and repeated some of the downstream analyses. In particular, we repeated these enrichments:

– Enrichment for different regulatory elements: Figure 3—figure supplement 1;

– Enrichment for CpGs reported to be associated with phenotypic traits and/or environmental exposures: Figure 3—figure supplement 1B;

– Heritability of methylation levels: Figure 4—figure supplement 1;

– Proportion of CpGs having meQTLs (methylation quantitative trait loci): Figure 4 –

figure supplement 2;

– Enrichment for CpGs with age variable methylation levels: Figure 5 – figure

supplement 1.

In addition, we also analyzed the distribution of the ICC for different types of CpGs:

– Probe reliability by CpG type (eQTM vs non-eQTM): Figure 1—figure supplement 4;

– Probe reliability by eQTM type (genetically determined eQTM vs not genetically determined eQTM): Figure 4—figure supplement 3;

– Probe reliability by eQTM type (age-shared eQTM vs age-specific eQTM): Figure 5—figure supplement 2.

These analyses are shown as supplemental figures and discussed briefly in the main text.

Finally in the Discussion section we have a specific comment on this issue (page 32 – line 761):

“As previously described (Sugden et al., 2020), CpG probes have different measurement error and thus different reliability and reproducibility. […] Moreover, the enrichments for active chromatin states were amplified and differences between inverse and positive eCpGs attenuated.”

3. Much of the paper is unnecessarily difficult to read. Many sections contain high levels of detail which could easily be summarized and the details lookoed up in provided tables or figures (e.g. "CpGs in eQTMs are enriched for blood active chromatin states and medium methylation levels" and "Annotating CpGs to the closest gene only partially captures eQTMs").

We appreciate the advice of the reviewer. We have simplified the text keeping the main messages and now details are only provided in tables and figures. In addition, only the results of the model adjusted for cellular composition are presented, which helps to keep the text more fluent.

4. The Discussion section contains a lot of awkward phrases making it difficult to read.

The manuscript has been reviewed by a native English speaker.

References:

GTEx consortium, The GTEx Consortium atlas of genetic regulatory effects across human tissues, Science (2020) Sep 11;369(6509):1318-1330. doi: 10.1126/science.aaz1776.

Sugden K, Hannon EJ, Arseneault L, Belsky DW, Corcoran DL, Fisher HL, Houts RM, Kandaswamy R, Moffitt TE, Poulton R, Prinz JA, Rasmussen LJH, Williams BS, Wong CCY, Mill J, Caspi A. Patterns of Reliability: Assessing the Reproducibility and Integrity of DNA Methylation Measurement. Patterns (N Y). 2020 May

8;1(2):100014. doi: 10.1016/j.patter.2020.100014. Epub 2020 Apr 23. PMID: 32885222; PMCID: PMC7467214.

GTEx consortium, The GTEx Consortium atlas of genetic regulatory effects across human tissues, Science (2020) Sep 11;369(6509):1318-1330. doi: 10.1126/science.aaz1776.

[Editors' note: further revisions were suggested prior to acceptance, as described below.]

Reviewer #2 (Recommendations for the authors):The authors replied to reviewers' comments well.Website is still PW protected.

We apologize for this issue with the web site. Summarized results of the eQTM catalogue are in the homepage of the website, which is open. However, according to the comment of the reviewer it seems they are not visible. We will amend this by moving them to another place on the website to gain visibility. In addition, the results are publicly available in Dryad (doi:10.5061/dryad.fxpnvx0t0).

Suggest defining autosomal in abstract and introduction.

We think autosomal chromosomes does not need a definition. However, for clarity, we already have added this comment in the Method section (line 308, page 14):

“Only CpGs in autosomal chromosomes (from chromosome 1 to 22) were tested.”

1. The section "Enrichment for age-variable eQTMs" is very confusing.a. Did you conduct EWAS of age in the cohort?b. Term "variable methylation is confusing". If you did EWAS, then call EWAS methylation.c. How many methylation CpGs did you test in each cohort?

We have tried to clarify this section. In this section, we did not run EWAS of age in HELIX, but compared the eCpGs we identified with CpGs whose methylation change from birth to childhood or adolescence, according to previous publications. We have modified the paragraph to clarify this point (page 28, line 656):

“To understand the association between changes in methylation and gene expression throughout life, first we evaluated whether eCpGs were enriched for CpGs whose methylation levels change from birth to childhood/adolescence according to the literature. To this end, we retrieved the CpGs that vary with age from two databases (RH et al., 2021; Xu et al., 2017)”

We agree with the reviewer that the term variable methylation can be confusing in this context. Therefore, we have changed the age-variable CpGs by CpG whose methylation levels change with age in the subheading (now is Enrichment for CpG whose methylation change with age), the paragraph and Figure 5 caption.